# Scalable Approximation Algorithms for $p$-Wasserstein Distance and Its Variants

**Nathaniel Lahn** [1]   **Sharath Raghvendra** [2]   **Emma Saarinen** [2]   **Pouyan Shirzadian** [3]

## Abstract

The $p$-Wasserstein distance measures the cost of optimally transporting one distribution to another, where the cost of moving a unit mass from $a$ to $b$ is the $p^{th}$ power of the ground distance $\mathrm{d}(a, b)$ between them. Despite its strong theoretical properties, its use in practice – especially for $p \geq 2$ – is limited due to two key challenges: sensitivity to noise and a lack of scalable algorithms. We identify noise sensitivity as a key reason why some existing approximation algorithms for $p = 1$ fail to generalize to $p \geq 2$ and then present new algorithms for approximating the $p$-Wasserstein distance and its variant. First, when $\mathrm{d}(\cdot, \cdot)$ is a metric, for any constant $p \geq 2$, we present a novel relative $O(\log n)$- approximation algorithm to compute the $p$-Wasserstein distance between any two discrete distributions of size $n$. The algorithm runs in $O(n^2 \log U \log \Delta \log n)$ time, where $\log U$ is the bit-length of the input probabilities and $\Delta$ is the ratio of the largest to the smallest pairwise distance. We use $p$ hierarchically well-separated trees to define a distance that approximates the $p$-Wasserstein cost within a factor of $O(\log n)$ and then present a simple primal-dual algorithm to compute the $p$-Wasserstein cost with respect to this distance. Second, due to the noise sensitivity of the $p$-Wasserstein distance, we show that existing combinatorial approaches require $\Omega(n^2/\delta^p)$ time to approximate the $p$-Wasserstein distance within an additive error of $\delta$. In contrast, we show that, for any arbitrary distance $\mathrm{d}(\cdot, \cdot)$, a recent noise-resistant variant of the $p$-Wasserstein distance, called the $p$-RPW distance, can be approximated in $O(n^2/\delta^3)$ time.

[*]Authors are listed Alphabetically   [1]Radford University [2]North Carolina State University [3]Virginia Tech. Correspondence to: N. Lahn <nlahn@radford.edu>, S. Raghvendra <skraghve@ncsu.edu>, E. Saarinen <ecsaarin@ncsu.edu>, P. Shirzadian <pshirzadian@vt.edu>.

*Proceedings of the 42$^{nd}$ International Conference on Machine Learning*, Vancouver, Canada. PMLR 267, 2025. Copyright 2025 by the author(s).

## 1. Introduction

Let $\mu$ and $\nu$ be discrete probability distributions supported on sets $A$ and $B$, respectively, with $|A| + |B| = n$. For each pair $(a, b) \in A \times B$, let $\mathrm{d}(a, b)$ denote the distance between $a$ and $b$. For simplicity, assume that the diameter of $A \cup B$ is 1, i.e., $\max_{(a,b) \in A \times B} \mathrm{d}(a, b) = 1$. A *transport plan* is a function $\sigma : A \times B \to \mathbb{R}_{\geq 0}$ that assigns a mass to each pair $(a, b)$ such that $\sum_{b \in B} \sigma(a, b) \leq \mu(a)$ and $\sum_{a \in A} \sigma(a, b) \leq \nu(b)$. Given a parameter $p \geq 1$, suppose the cost of moving a unit of mass from a point $a \in A$ to a point $b \in B$ is given by $\mathrm{d}(a, b)^p$. The $p$-Wasserstein cost of any transport plan $\sigma$ between $\mu$ and $\nu$ is defined as

$$w_p(\sigma) := \left( \sum_{a \in A, b \in B} \sigma(a, b) \times \mathrm{d}(a, b)^p \right)^{1/p}.$$

In the $\alpha$-partial $p$-Wasserstein problem, we wish to compute the transport plan $\sigma_\alpha^*$ that transports $\alpha$ fraction of the mass and has the smallest $p$-Wasserstein cost. We refer to this cost as the $\alpha$-partial $p$-Wasserstein distance and denote it by $W_{p,\alpha}(\mu, \nu) = w_p(\sigma_\alpha^*)$. The $p$-*Wasserstein distance*, denoted by $W_p(\mu, \nu)$, is equal to the cost $w_p(\sigma_1^*)$.

A standard approach to estimating the $p$-Wasserstein distance between $\mu$ and $\nu$ is to draw $n$ samples from both and construct empirical distributions $\mu_n$ and $\nu_n$ by assigning a mass of $1/n$ to each sampled point. Computing the $p$-Wasserstein distance between these empirical distributions corresponds to a well-known problem in combinatorial optimization: the *assignment problem*. In this setting, the optimal transport plan is a *matching* – a set of $n$ vertex-disjoint edges, each carrying a mass of $1/n$.

The $p$-Wasserstein distance is effective in capturing geometric similarity between distributions, especially when $p \geq 2$. When the ground distance $\mathrm{d}(\cdot, \cdot)$ is a metric, the $p$-Wasserstein distance is also a metric. Furthermore, when the diameter of the supports is bounded, the empirical $p$-Wasserstein distance converges to the true distance as the sample size increases (Fournier and Guillin, 2015). These favorable theoretical properties have led to its widespread adoption in applications across machine learning (Chang et al., 2023; Chuang et al., 2022), computer vision (Backurs et al., 2020; Lai et al., 2022), and natural language processing (Alvarez-Melis and Jaakkola, 2018; Yurochkin et al.,

2019). However, despite its strengths, the $p$-Wasserstein distance (for $p \geq 2$) faces two significant limitations in practice: *sensitivity to noise* and *lack of scalable algorithms*.

**Noise sensitivity:** A major drawback of the $p$-Wasserstein distance is its sensitivity to small amounts of noise. Specifically, a noise mass of $\delta$ can increase the $p$-Wasserstein distance by as much as $\delta^{1/p}$. For example, a $1\%$ noise ($\delta = 0.01$) can increase the 2-Wasserstein distance by more than $10\%$ (or $0.1$) (Raghvendra et al., 2024). This makes the $p$-Wasserstein distance less reliable for real-world datasets which often contain noise or outliers. To address this limitation, Raghvendra et al. (2024) introduced the $(p, k)$-Robust Partial Wasserstein ($(p, k)$-RPW) distance. It is defined as the smallest $\varepsilon$ such that $W_{p,(1-\varepsilon)}(\mu, \nu) \leq k\varepsilon$. This formulation generalizes several classical metrics including total variation, the $p$-Wasserstein, and the well-known Lèvy-Prokhorov distance (Prokhorov, 1956). Importantly, the $(p, k)$-RPW distance is provably more robust: modifying a $\delta$ mass in the distribution alters the $(p, k)$-RPW distance by at most $\delta$, in contrast to the potentially much larger distortion in the standard $p$-Wasserstein case.

**Exact Algorithms:** Exact algorithms for computing the $p$-Wasserstein distance take $\tilde{O}(n^3)$ time (Orlin, 1988)[1]. For the case where the precision of costs and probabilities are bounded, one can use interior point methods to improve the execution time to $O(n^{2+o(1)})$ (Chen et al., 2022); however, these algorithms are complex and, without significant simplifications, it is unlikely that these methods will lead to a usable implementation for machine learning applications. Similarly, for distances that admit a dynamic weighted bichromatic closest pair (BCP) data structure with a query and update time of $\Phi(n)$, one can improve the execution time of exact algorithms to $\tilde{O}(n^2\Phi(n))$. Unfortunately, the only known BCP data structures are in the two-dimensional setting and the data structures are based on sophisticated techniques that do not have an implementation (Eppstein, 1995; Chan, 2020). Due to a lack of efficient, usable exact algorithms, researchers have focused on designing relative and additive approximation algorithms.

**Relative Approximations:** An $\alpha$-relative approximation algorithm (or simply an $\alpha$-approximation) returns a transport plan whose $p$-Wasserstein cost is at most $\alpha$ times the optimal value. When the ground distance $\mathrm{d}(\cdot, \cdot)$ is a metric and $p = 1$, Charikar (2002) presented an $O(n^2)$ time $O(\log n \log \log n)$-approximation algorithm. The key idea is to approximate the original metric with a tree metric that preserves the 1-Wasserstein distance within a multiplicative factor of $O(\log n \log \log n)$. Once embedded into the tree, the 1-Wasserstein distance under the tree metric can be computed efficiently using a simple greedy algorithm. However, this approach does not generalize to $p \geq 2$, since the

tree metric cannot approximate the 2-Wasserstein distance within a bounded factor.

Subsequent work by Agarwal and Sharathkumar (2014) presented an algorithm to find an $O(1/\delta^{0.631})$-approximation of the 1-Wasserstein distance in $O(n^{2+\delta})$ time, which was later improved by Sherman (2017) to a $(1 + \varepsilon)$ approximation algorithm in $\tilde{O}(n^2/\varepsilon^2)$ time. These algorithms rely on the assumption that the cost function is a metric – a property that holds when $p = 1$, but fails for $p \geq 2$ since the costs are $\mathrm{d}(\cdot, \cdot)^p$, which do not satisfy the triangle inequality. As a result, these methods do not extend to higher values of $p$.

For $p = 1$, significantly faster algorithms exist in geometric settings (e.g., low-dimensional Euclidean space); see, for instance, (Agarwal et al., 2022; Indyk, 2007; Sharathkumar and Agarwal, 2012a;b; Agarwal and Varadarajan, 2004; Fox and Lu, 2020). Owing to their simplicity, some of these algorithms have since been adapted to the design of data structures and algorithms for more complex tasks in the Wasserstein space; see, for instance, nearest neighbor searching (Backurs et al., 2020; Andoni et al., 2018) and barycenter computation (Agarwal et al., 2025). In contrast, for $p \geq 2$, there are no fast approximation algorithms known that work for any metric space. The only existing fast approximation algorithms are restricted to specific settings, for instance two-dimensional Euclidean settings (Lahn and Raghvendra, 2019; 2021). In this paper, we address the following major open challenge:

*For any integer $p \geq 2$ and any metric $\mathrm{d}(\cdot, \cdot)$, can we design an $\tilde{O}(n^2)$ time $O(\log n)$-relative approximation algorithm to compute the $p$-Wasserstein distance between two discrete distributions?*

**Additive Approximations:** A transport plan $\sigma$ is a $\delta$-*additive approximation*, or simply $\delta$-*close*, if $w_p(\sigma) \leq w_p(\sigma^*) + \delta$. There are several simple and highly parallelizable quadratic algorithms for computing additive approximations for any cost matrix (Cuturi, 2013; Altschuler et al., 2017; 2019; Lin et al., 2019; Dvurechensky et al., 2018; Jambulapati et al., 2019; Lahn et al., 2019; 2023). However, the best-known algorithms run in $\tilde{O}(n^2/\delta^p)$ time, which we show is likely difficult to improve upon. Note that the execution times of these algorithms increase with $p$ and do not converge for $p = \infty$.

## 1.1. Our Results

In this paper, we identify noise sensitivity as a key barrier preventing the extension of scalable relative and additive approximation algorithm for the 1-Wasserstein distance to the case where $p \geq 2$, and we present new algorithms that overcome this challenge.

**Relative Approximation:** First, for any finite integer $p \geq 2$, we present an $\tilde{O}(n^2)$ time $O(\log n)$-relative approxima-

---

[1] $\tilde{O}(\cdot)$ hides $\mathrm{poly} \log n$ factors in the running time

tion algorithm for the $p$-Wasserstein distance. We are not aware of any sub-linear approximation algorithm for the $p$-Wasserstein problem that runs in near-quadratic time.

**Theorem 1.1.** *Given two discrete distributions $\mu$ and $\nu$ defined over point sets $A$ and $B$, respectively, where $|A| + |B| = n$, a metric $\mathrm{d}(\cdot,\cdot)$, and a parameter $p \geq 1$, there exists an $O(\log n)$-approximation algorithm to compute the $p$-Wasserstein distance that runs in $O(n^2 \log U \log \Delta \log n)$ time, where $\log U$ is the number of bits required to represent the probability values and $\Delta$ is the spread of $A \cup B$.*

Previous relative approximation algorithms for the 1-Wasserstein distance rely on embedding the ground metric $\mathrm{d}(\cdot,\cdot)$ into a tree metric by using a hierarchically well-separated tree (HST) (Charikar, 2002; Kleinberg and Tardos, 2002). Embedding a metric into a tree metric introduces distortion – a form of noise in the edge costs – which is tolerable for $p = 1$ but problematic for $p \geq 2$ due to the increased sensitivity of the $p$-Wasserstein distance to such perturbations. As a result, the $p$-Wasserstein cost computed over the tree metric can deviate significantly from the true cost, yielding unbounded approximation guarantees. To overcome this limitation, we propose a new approach that reduces the impact of distortion by considering multiple tree embeddings. Specifically, we construct $p$ different HST embeddings and, for each pair of points, use the minimum edge cost across these embeddings. We show that this strategy yields a more stable distance, and prove that the optimal $p$-Wasserstein distance computed using these modified edge costs provides an $O(\log n)$-approximation to the true $p$-Wasserstein distance.

Unlike the case of the 1-Wasserstein distance, a simple greedy algorithm does not yield an optimal solution under our new distance. Instead, we adopt a primal-dual framework to compute the optimal transport plan. In general, primal-dual algorithms perform $O(n)$ graph search steps to identify so-called augmenting paths, with each search typically taking $O(n^2)$ time—resulting in an overall time complexity of $O(n^3)$. To improve efficiency, we leverage the structure of the $p$ tree embeddings and design a dynamic BCP data structure to accelerate the search process. Using this data structure, we conduct each graph search in $O(n \log \Delta \log n)$ time, where $\Delta$ is the spread, which is the ratio between the largest and smallest nonzero edge costs. This results in a total runtime of $O(n^2 \log \Delta \log n)$ for computing the optimal solution.

We also provide an implementation of our algorithm and empirically analyze the efficiency and accuracy of the solution produced by our algorithm on stochastic data sets.

**Additive Approximation:** Using the sensitivity of $p$-Wasserstein distance to noise, we show that any algorithm that approximates the $p$-Wasserstein distance within an additive factor of $\delta$ and runs in better than $O(n^2/\delta^p)$ time

can be used to determine whether an unweighted bipartite graph has a perfect matching in faster than $n^{2.5}$ time. While there are several theoretically almost-quadratic algorithms (e.g. (Chen et al., 2022)) for finding a perfect matching, the fastest practical algorithm remains the classic Hopcroft-Karp algorithm Hopcroft and Karp (1973), which runs in $O(n^{2.5})$ time. This reduction suggests a barrier to designing efficient and implementable algorithms for computing the $p$-Wasserstein distance faster than $O(n^2/\delta^p)$ time. This computational hardness extends to robust variants as well, including the $\alpha$-partial $p$-Wasserstein distance (Chapel et al., 2020) and the $\lambda$-ROBOT distance (Mukherjee et al., 2021).

Interestingly, we show that the $(p, k)$-RPW distance can be approximated in $O(n^2/\delta^3)$ time, for any $p \geq 1$ and fixed constant $k$, and this runtime is independent of $p$. Thus, the approximate $(p, k)$-RPW distance overcomes both major limitations of the $p$-Wasserstein distance: it is robust to noise and supports significantly more scalable algorithms.

**Theorem 1.2.** *Given two discrete distributions $\mu$ and $\nu$ defined over point sets $A$ and $B$, respectively, where $|A| + |B| = n$, a metric $\mathrm{d}(\cdot,\cdot)$, and parameters $p \geq 1$, $\delta > 0$, and any fixed constant $k > 0$, there exists a $\delta$-additive approximation algorithm for the $(p, k)$-RPW distance that runs in $O(n^2/\delta^3)$ time.*

Our improved algorithm for approximating the $(p, k)$-RPW distance is motivated by a key observation: augmenting-path-based optimal transport algorithms tend to transport most of the mass quickly, with the majority of the runtime spent on routing a small residual fraction (Hopcroft and Karp, 1973; Lahn et al., 2019; Lahn and Raghvendra, 2019). For example, using the classical Hopcroft-Karp algorithm (Hopcroft and Karp, 1973) for bipartite perfect matching, a matching of size $(1 - \delta)n$ can be computed in $O(n^2/\delta)$ time for any constant $\delta > 0$, whereas completing the remaining $\delta n$ edges requires $n^{2.5}$ time. A similar behavior also holds for several combinatorial algorithms that approximate the optimal transport cost.

In contrast, to approximate the $(p, k)$-RPW distance $\varepsilon$ between distributions $\mu$ and $\nu$ up to an additive error of $\delta$, it suffices to compute a transport plan that moves at least $(1 - \varepsilon - \delta)$ mass with a total cost at most $\varepsilon + \delta$. We leverage this relaxation to design a faster approximation algorithm by adapting the LMR algorithm (Lahn et al., 2019). Specifically, we find a transport plan that deviates from the optimal in both cost and transported mass by at most $\delta$, thereby approximating the RPW distance efficiently. We also provide an implementation of our algorithm. Experiments (included in Appendix C) suggest that our algorithm outperforms the algorithm by Raghvendra et al. (2024), especially for larger values of $p$ and smaller values of $\delta$.

## 2. A Tree Based Distance

In this section, we present a novel distance function based on $p$ independently constructed hierarchically well-separated trees (HSTs) that approximates the ground metric $d(\cdot, \cdot)$. We prove that the $p$-Wasserstein distance with respect to our tree-based distance is, in expectation, an $O(\log n)$ approximation of the $p$-Wasserstein distance with respect to $d(\cdot, \cdot)$.

We begin by reviewing HSTs and their key properties as established by Fakcharoenphol et al. (2003) in Section 2.1. In Section 2.2, we formally define our new distance function and demonstrate its approximation guarantees. Then, in Section 2.3, we present a dynamic weighted bichromatic closest pair (BCP) data structure for our tree-based distance. Finally, in Section 3, we use this BCP data structure to design a near-quadratic time algorithm for computing the $p$-Wasserstein distance with respect to our tree-based distance.

### 2.1. Hierarchically Well-Separated Trees

Let $(X, d)$ be a metric space. Consider a partition $P = \{C_1, \ldots, C_t\}$ of $X$ into clusters, where each point in $X$ belongs to exactly one cluster in $P$. For each cluster $C \in P$, let $\text{Diam}(C)$ denote the diameter of $C$, and let $X_C$ denote the subset of $X$ contained in $C$. For each point $x \in X$ and radius $\delta > 0$, define a ball centered at $x$ and radius $\delta$, denoted by $B(x, \delta)$, as the points of $X$ that are at a distance no more than $\delta$ from $x$, i.e., $B(x, \delta) := \{u \in X \mid d(x, u) \leq \delta\}$. For any integer $j \geq 0$, define $k_j^x := |B(x, 2^{-j})|$ as the number of points of $X$ that are at a distance at most $2^{-j}$ from $x$.

Fakcharoenphol et al. (2003) presented a randomized algorithm that, given a metric space $(X, d)$ with unit diameter and spread $\Delta$, constructs a hierarchically well-separated tree (HST) $T$ over the points in $X$ in $O(n^2)$ time. The root of the tree corresponds to all points in $X$, the children of each cluster $C$ form a partition of the subset $X_C$, and every point in $X$ forms a leaf cluster in the tree. The level of a cluster $C$ is defined as its distance (in number of edges) from the root: the root is at level 0, and the levels increase as we descend the tree. Their construction guarantees two properties:

(H1) For any cluster $C$ at level $j$, $\text{Diam}(X_C) \leq 2^{-j}$.

(H2) For any pair of points $u, v \in X$, the probability that the least common ancestor of the pair $(u, v)$ is at a level at most $j$ is at most $\frac{d(u,v)}{2^{-j-1}} \times \Gamma(u, v, j)$, where $\Gamma(u, v, j) := H_{k_j^u} + H_{k_j^v} - H_{k_{j+2}^u} - H_{k_{j+2}^v}$; here, $H_k$ is the $k$th Harmonic number.

The tree $T$ has a height $O(\log \Delta)$. Moreover, every point $x \in X$ participates in each of the $O(\log \Delta)$ clusters that are ancestors of $x$ in $T$.

### 2.2. Distance Function

Let $\mu$ and $\nu$ be two discrete distributions supported on point sets $A$ and $B$, with $|A| + |B| = n$, and let $d(\cdot, \cdot)$ be the ground metric. In this section, we introduce an HST-based distance function $d_{\mathcal{T}}(\cdot, \cdot)$ and show that the $p$-Wasserstein distance with $d_{\mathcal{T}}(\cdot, \cdot)$ as the ground distance would be an $O(\log n)$-approximation of the $p$-Wasserstein distance with $d(\cdot, \cdot)$ as the ground distance.

Let $\mathcal{T} = \{T_1, \ldots, T_p\}$ be a set of $p$ independently constructed HSTs over the metric space $(A \cup B, d)$. For any pair of points $(a, b) \in A \times B$ and each tree index $i \in [1, p]$, let cluster $\text{C}_i(a, b)$ denote the least common ancestor of $a$ and $b$ in the tree $T_i$ and let $\text{lev}_i(a, b)$ denote the level of the cluster $\text{C}_i(a, b)$ in the tree. Recall that the diameter of the cluster at level $j$ is at most $2^{-j}$; hence, $2^{-\text{lev}_i(a,b)}$ is an upper bound on $d(a, b)$. We define the *tree distance* between $a$ and $b$ as

$$d_{\mathcal{T}}(a, b) := \min_{i \in [1,p]} \{2^{-\text{lev}_i(a,b)}\}.$$

This definition takes the best (i.e., tightest) estimate among all the $p$ trees as an upper bound for $d(a, b)$. The next lemma relates $d_{\mathcal{T}}(a, b)$ to the ground distance $d(a, b)$.

**Lemma 2.1.** *For any pair of points* $(a, b) \in A \times B$, $d_{\mathcal{T}}(a, b)^p \geq d(a, b)^p$ *and* $\mathbb{E}[d_{\mathcal{T}}(a, b)^p] = O((4 \log n)^p d(a, b)^p)$.

*Proof.* Let $j = \arg\min_{i \in [1,p]} \{2^{-\text{lev}_i(a,b)}\}$, i.e. $d_{\mathcal{T}}(a, b) = 2^{-\text{lev}_j(a,b)}$. Recall that $\text{C}_j(a, b)$ denotes the least common ancestor of $a$ and $b$ in $T_j$. Then,

$$d_{\mathcal{T}}(a, b)^p = (2^{-\text{lev}_j(a,b)})^p \geq \text{Diam}(\text{C}_j(a, b))^p \geq d(a, b)^p,$$

where the second inequality holds from (H1) and the last inequality holds since $a, b \in \text{C}_j(a, b)$. Let $h$ denote the maximum height of the HSTs in $\mathcal{T}$. For any level $j \in [0, h]$,

$$\Pr[\text{lev}(a, b) = j] \leq \Pr[\text{lev}_i(a, b) \leq j, \forall i \in [1, p]]$$
$$= \prod_{i=1}^{p} \Pr[\text{lev}_i(a, b) \leq j]$$
$$\leq \frac{d(a, b)^p \cdot \Gamma(a, b, j)^p}{(2^{-j-1})^p}$$
$$\leq \frac{d(a, b)^p \cdot (2H_n)^{p-1} \cdot \Gamma(a, b, j)}{2^{-p(j+1)}},$$

where the last inequality holds since $k_j^u \in [1, n]$ and $\Gamma(a, b, j) \leq 2H_n$. We can bound the expected distortion of the tree distance $d_{\mathcal{T}}(a, b)$ by summing over all levels from

0 to $h$ as follows.

$$
\begin{aligned}
\mathbb{E}\left[\mathrm{d}_{\mathcal{T}}(a,b)^p\right] &= \sum_{j=0}^{h} \Pr\left[\mathrm{lev}(a,b)=j\right]\cdot(2^{-j})^p \\
&\leq \sum_{j=0}^{h} \frac{\mathrm{d}(a,b)^p\cdot(2H_n)^{p-1}\cdot\Gamma(a,b,j)}{2^{-p(j+1)}}\cdot 2^{-pj} \\
&= 2^p\cdot(2H_n)^{p-1}\cdot\mathrm{d}(a,b)^p\sum_{j=0}^{h}\Gamma(a,b,j) \\
&\leq 2^p\cdot(2H_n)^{p-1}\cdot\mathrm{d}(a,b)^p\times 4H_n \qquad (1) \\
&= O((4\log n)^p\times\mathrm{d}(a,b)^p),
\end{aligned}
$$

where Inequality (1) holds since

$$
\begin{aligned}
\sum_{j=0}^{h}\Gamma(a,b,j) &= \sum_{j=0}^{h} H_{k_j^a} + H_{k_j^b} - H_{k_{j+2}^a} - H_{k_{j+2}^b} \\
&= H_{k_0^a} + H_{k_0^b} + H_{k_1^a} + H_{k_1^b} \\
&\quad - H_{k_{h+1}^a} - H_{k_{h+1}^b} - H_{k_{h+2}^a} - H_{k_{h+2}^b} \\
&\leq 4H_n.
\end{aligned}
$$

$\square$

Let $\sigma^*$ (resp. $\sigma_{\mathcal{T}}^*$) be the transport plan between $\mu$ and $\nu$ that minimizes the $p$-Wasserstein cost with respect to the distances $\mathrm{d}(\cdot,\cdot)$ (resp. $\mathrm{d}_{\mathcal{T}}(\cdot,\cdot)$). For any transport plan $\sigma'$, let $w_{\mathcal{T},p}(\sigma')$ denote the $p$-Wasserstein cost of $\sigma'$ under the tree distances $\mathrm{d}_{\mathcal{T}}(\cdot,\cdot)$. Using linearity of expectation, we bound the expected value $\mathbb{E}[w_{\mathcal{T},p}(\sigma_{\mathcal{T}}^*)^p]$ by $O((4\log n)^p w_p(\sigma^*)^p)$. Combining it with Jensen's inequality, we have the following lemma.

**Lemma 2.2.** $\mathbb{E}[w_{\mathcal{T},p}(\sigma_{\mathcal{T}}^*)] = O((\log n)w_p(\sigma^*))$.

Next, we describe a simple dynamic data structure for maintaining a weighted bichromatic closest pair between two subsets $U \subseteq B$ and $V \subseteq A$. In Section 3, we use this data structure in a primal-dual algorithm for computing an approximation of the $p$-Wasserstein distance.

### 2.3. Dynamic Bichromatic Closest Pair Data Structure

Let $\mathrm{w}(u)$ denote the weight associated with point $u$. For any pair of points $a \in A$ and $b \in B$, define the *weighted distance* between them as:

$$
\mathrm{d}_{\mathcal{T},\mathrm{w}}(a,b) := \mathrm{d}_{\mathcal{T}}(a,b)^p - \mathrm{w}(a) - \mathrm{w}(b).
$$

Given two sets of points $U \subseteq B$ and $V \subseteq A$, the *weighted bichromatic closest pair* between $U$ and $V$ is the pair minimizing the weighted distance, i.e.,

$$
\arg\min_{u\in U, v\in V}\{\mathrm{d}_{\mathcal{T}}(u,v)^p - \mathrm{w}(u) - \mathrm{w}(v)\}.
$$

We present a dynamic data structure that maintains this pair and supports insertions and deletions in $U$ and $V$ in $O(p\log n\log\Delta)$ time. For each tree $T_i \in \mathcal{T}$ and each cluster $C \in T_i$, let $U_C$ (resp. $V_C$) denote the subset of $U$ (resp. $V$) that participates in $C$. Let $\mathrm{lev}(C)$ denote the level of $C$ in tree $T_i$. The following key lemma is instrumental in the design of our data structure.

**Lemma 2.3.** *Suppose $(a^*, b^*) \in V \times U$ is the weighted bichromatic closest pair between $V$ and $U$. Then,*

$$
\begin{aligned}
&\mathrm{d}_{\mathcal{T},\mathrm{w}}(a^*, b^*) \\
&= \min_{T_i\in\mathcal{T}, C\in T_i}\left\{2^{-\mathrm{lev}(C)\cdot p} - \max_{b\in U_C}\mathrm{w}(b) - \max_{a\in V_C}\mathrm{w}(a)\right\}.
\end{aligned}
$$

**Data Structure:** For each tree $T_i \in \mathcal{T}$ and each cluster $C \in T_i$, define the *candidate closest pair* for cluster $C$ as the pair $(a_C, b_C) \in V_C \times U_C$ that minimizes $2^{\mathrm{lev}(C)\cdot p} - \max_{b\in U_C}\mathrm{w}(b) - \max_{a\in V_C}\mathrm{w}(a)$. To maintain the candidate closest pair of $C$, our data structure stores two max-heaps for each cluster $C$: $\mathrm{Heap}_C^V$ for points in $V_C$ and $\mathrm{Heap}_C^U$ for points in $U_C$, with weights as keys. The candidate closest pair for $C$ is simply given by the roots of these heaps.

We maintain a global min-heap $\mathrm{GlobalHeap}$ that stores the candidate pair for each cluster $C$, with their weighted distance as key. From Lemma 2.3, the weighted bichromatic closest pair corresponds to the minimum entry in $\mathrm{GlobalHeap}$.

To insert (resp. delete) a point $b \in B$ into (resp. from) the set $U$, we proceed as follows: for each tree $T_i \in \mathcal{T}$ and every cluster $C$ along the path from the leaf containing $b$ to the root of $T_i$, we insert $b$ into (resp. remove $b$ from) the heap $\mathrm{Heap}_C^U$. If the operation alters the candidate closest pair for any such cluster, we update its entry in the global min-heap $\mathrm{GlobalHeap}$. An identical procedure is used to insert or delete a point in the set $V$ by updating the corresponding heaps $\mathrm{Heap}_C^V$.

**Analysis:** Building all $p$ HSTs takes $O(pn^2)$ time. The current bichromatic closest pair is always at the root of $\mathrm{GlobalHeap}$ and can be queried in $O(1)$ time. Each point $v \in A \cup B$ belongs in $O(\log\Delta)$ clusters per HST, or $O(p\log\Delta)$ clusters overall. Inserting or deleting a point in either $U$ or $V$ requires updating the corresponding heaps in all these $O(p\log\Delta)$ *affected* clusters, recomputing candidate pairs, and updating $\mathrm{GlobalHeap}$. This takes $O(p\log n\log\Delta)$ time per update.

**Lemma 2.4.** *We can build a data structure $\mathcal{D}$ in $O(pn^2)$ time that stores weighted point sets $U$ and $V$, returns the bichromatic closest pair in $O(1)$ time, and allows for insertion and deletion of points to $U \cup V$ in $O(p\log n\log\Delta)$ time.*

**Remark:** The space complexity of the data structure is $O(pn\log\Delta)$. This can be reduced to $O(pn)$ by optimizing

the heap storage: instead of storing all points in $U_C$ and $V_C$, we store only one representative per child cluster (the point with the maximum weight) in the non-leaf clusters.

# 3. An $O(\log n)$-Approximation Algorithm for the Optimal Transport

Given a dynamic weighted bichromatic closest pair (BCP) data structure with a query/update time of $\Phi(n)$, the optimal $p$-Wasserstein distance between two distributions can be computed in $O(n^2\Phi(n)\log U)$ time; here $U$ is the number of bits required to store the probability associated with each point. For completeness, we briefly describe the classical BCP-based algorithm for the assignment problem as introduced by (Vaidya, 1989).

## 3.1. Primal-Dual Framework

Consider a complete bipartite graph $G(A \cup B, A \times B)$ defined on sets $A$ and $B$. A *matching* $M \subseteq A \times B$ is a set of vertex-disjoint edges. The *size* of a matching is the number of edges it contains. A vertex $v$ is called *free* (with respect to $M$) if no edges of $M$ are incident on $v$. Let $A_F^M$ and $B_F^M$ be the set of free vertices in $A$ and $B$, respectively; we drop the superscript $M$ when the matching is clear from the context. An *alternating path* is a path that alternates between edges in $M$ and those that are not in $M$, and an *augmenting path* is one that starts and ends at a free vertex. We *augment* $M$ along such a path $P$ by setting $M \leftarrow M \oplus P$, where edges in $P \cap M$ are removed and those in $P \setminus M$ are added, increasing the size of matching $M$ by 1.

We associate a dual weight $y(v)$ with each vertex $v \in A \cup B$, and say that the matching $M$ and dual weights $y(\cdot)$ is *dual feasible* if:

$$y(a) + y(b) \leq \mathrm{d}(a,b)^p \qquad \text{for all } (a,b) \notin M, \quad (2)$$
$$y(a) + y(b) = \mathrm{d}(a,b)^p \qquad \text{for all } (a,b) \in M. \quad (3)$$

These are the classical feasibility conditions for the assignment problem.

**Lemma 3.1.** *If a perfect matching $M$ along with a set of dual weights is dual feasible, then $M$ is a minimum-cost perfect matching.*

We now describe how a dynamic bichromatic closest pair data structure can be used to implement the Hungarian algorithm in $O(n^2\Phi(n))$ time and how it applies to computing the $p$-Wasserstein distance.

## 3.2. Hungarian Algorithm

To find augmenting paths, the Hungarian algorithm constructs a *residual graph* $G_M$. The vertex set for $G_M$ is $A \cup B$, a *source* vertex $s$ and a *sink* vertex $t$. The edge set

of $G_M$ is defined as follows: For each $(a,b) \in A \times B$, if $(a,b) \notin M$, add a directed edge from $b$ to $a$; if $(a,b) \in M$, add a directed edge from $a$ to $b$. We set the cost of any edge between $a$ and $b$, regardless of direction, to be its *slack*, which is given by $\mathrm{d}(a,b)^p - y(a) - y(b)$. We also add a zero-cost edge directed from the source vertex to every vertex in $B_F$ and add a zero-cost edge directed from every vertex in $A_F$ to $t$.

The algorithm starts with an empty matching $M$ and initializes all dual weights $y(v) \leftarrow 0$ for all $v \in A \cup B$. It performs a sequence of Hungarian Search procedures each of which augments the matching while maintaining dual feasibility.Each Hungarian Search increases the size of $M$ by 1, and after $n$ such steps, we obtain a minimum-cost perfect matching.

**Hungarian Search Procedure:** The Hungarian search procedure consists of the following steps:

- Execute Dijkstra's shortest path algorithm with $s$ as the source on $G_M$. For any vertex $v$, let $P_v$ denote the shortest path from $s$ to $v$, and let $\ell_v$ be the total cost of $P_v$, using slacks as edge costs.

- For any $b \in B$, if $\ell_b < \ell_t$, set $y(b) \leftarrow y(b) - \ell_b + \ell_t$

- For any $a \in A$, if $\ell_a < \ell_t$, set $y(a) \leftarrow y(a) - \ell_t + \ell_a$

- Let $P$ be a path obtained by removing $s$ and $t$ from $P_t$. Augment $M$ along $P$.

Note that all steps in the Hungarian search except the execution of Dijkstra's algorithm take $O(n)$ time. We present an efficient $O(n\Phi(n))$ implementation of Dijkstra's shortest path algorithm on $G_M$.

**Efficient Dijkstra's shortest path algorithm:** Recollect, that Dijkstra's algorithm incrementally builds a *shortest path tree* rooted at the source $s$. Let $U \subseteq B$ be the points already added to the shortest path tree, and let $V \subseteq A$ be those not yet added. Initially, $U = B_F$ and $V = A$ the points in $B_F$. For each $b \in U$, $\ell_b = 0$ and $\mathrm{w}(b) = y(b) - \ell_b$. For each $a \in V$, assign $\mathrm{w}(a) = y(a)$. Build a BCP structure on $(U, V)$ using these weights. In each iteration of the algorithm, select

$$(a,b) \quad = \quad \arg\min_{a' \in V, b' \in U}\{s(a',b') + \ell_{b'}\}. \quad (4)$$

This is equivalent to

$$\begin{aligned}
s(a',b') + \ell_{b'} \quad &= \quad \mathrm{d}(a',b')^p - y(a') - (y(b') - \ell_{b'}) \\
&= \quad \mathrm{d}(a',b')^p - \mathrm{w}(a') - \mathrm{w}(b').
\end{aligned}$$

Thus, the pair minimizing equation 4 is exactly the BCP between $U$ and $V$ and can be found in $O(1)$ time.

Once the pair $(a, b)$ is selected, the algorithm will remove $a$ from $V$, add it to the shortest path tree, and set $\ell_a = \ell_b + s(a, b)$. If $a$ is free, we have found an augmenting path. Otherwise, let $b'$ be the vertex matched with $a$ in $M$. Set $\ell_{b'} = \ell_a$, $\mathrm{w}(b') = y(b') - \ell_{b'}$, and add $b'$ to $U$.

Each iteration of Dijkstra's deletes one vertex from $V$ and adds at most one vertex to $U$. The algorithm terminates with an augmenting path in at most $n$ iterations.

**Time Complexity:** Each iteration of Dijkstra's algorithm during the Hungarian Search adds at most one vertex to $U$ and removes one vertex from $V$. Since there are at most $n$ such iterations, the total number of insertions and deletions to the BCP data structure is $O(n)$. Given that each insertion and deletion takes $O(p \log \Delta \log n)$ time, the total execution time for the Hungarian search is $O(pn \log \Delta \log n)$. As the assignment problem requires $n$ such searches to compute a perfect matching, the overall complexity is $O(pn^2 \log \Delta \log n)$.

For the optimal transport problem, when the probabilities associated with each node can be represented using $O(\log U)$ bits, (Atkinson and Vaidya, 1995) showed that $O(n \log U)$ Hungarian searches are sufficient to find the optimal transport. Therefore the $p$-Wasserstein distance between two distributions $\mu$ and $\nu$ under the distance $\mathrm{d}(\cdot, \cdot)$ can be found in $O(pn^2 \log U \log \Delta \log n)$ time, where $\log U$ is the number of bits required to represent the probability at every point in the support of the distributions.

**Remark:** Sharathkumar and Agarwal (2012a) demonstrated that the optimal solution for the assignment problem can be computed efficiently by combining the BCP data structure with the cost-scaling framework. Their algorithm performs $\tilde{O}(n^{3/2} \log(nC))$ BCP queries, where $\log C$ is the number of bits needed to represent the edge costs. This approach can be integrated with our BCP data structure in a straightforward way to compute an $O(\log n)$-approximation to the $p$-Wasserstein distance. The overall execution time of this algorithm is dominated by the construction of $p$ HSTs and is $\tilde{O}(pn^2 + pn^{3/2} \log^2 \Delta)$.

# 4. Additive Approximation for Wasserstein Distance and Its Variants

In this section, we show that designing fast additive approximation algorithms for the $p$-Wasserstein distance is challenging due to its sensitivity to noise. In contrast, we show that a robust variant of the $p$-Wasserstein distance called the $(p, k)$-RPW distance can be approximated efficiently.

Consider the problem of determining if an unweighted bipartite graph $G(A \cup B, E)$ has a perfect matching. All sub-$n^{2.5}$ time algorithms for this problem are either matrix-multiplication-based approaches or sophisticated interior point methods. Thus, the best-known simple, implementable algorithm to determine if there is a perfect matching in $G$ is by Hopcroft and Karp (1973) and takes $\Theta(n^{2.5})$ time. Obtaining a simple, implementable algorithm with a sub-$n^{2.5}$ execution time remains a major open question. In Section 4.1, we present a reduction (similar to the one presented by (Blanchet et al., 2018)) demonstrating that any algorithm $\mathbb{A}$ that produces a $\delta$-additive approximation of the $p$-Wasserstein distance in sub-$n^2/\delta^p$ time can be adapted to achieve a sub-$n^{2.5}$ algorithm for finding a perfect matching, given a clever selection for the error parameter $\delta$.

## 4.1. Reduction

Given an instance of the perfect matching problem on the dense graph $G(A \cup B, E)$, with $|A| = |B| = n$, we convert $G$ to an instance of the $p$-Wasserstein problem as follows: Create distributions $\mu$ and $\nu$ by assigning a mass of $1/n$ for each point of $A$ and $B$. For each pair $(a, b) \in A \times B$, assign a distance $\mathrm{d}(a, b) = 0$ if $(a, b) \in E$ and a distance $\mathrm{d}(a, b) = 1$ otherwise. Now, given this instance of $p$-Wasserstein problem, we use any $\delta$-additive approximation algorithm $\mathbb{A}$ with running time expressed as $T(n, \delta, p)$ to compute a transport plan $\sigma$ and $p$-Wasserstein cost $w_p(\sigma)$ such that $w_p(\sigma) \leq w_p(\sigma^*) + \delta$, where $\sigma^*$ is an optimal transport plan. The optimal $p$-Wasserstein cost is equal to $((n - |M^*|)/n)^{1/p}$, where $M^*$ is the max-cardinality matching from our original graph $G$. Therefore, $\mathbb{A}$ returns a plan with $p$-Wasserstein cost at most $((n - |M^*|)/n)^{1/p} + \delta$. Note that if a perfect matching exists in $G$, then our total transport cost must be $(w_p(\sigma))^p \leq \delta^p$.

We convert the transport plan $\sigma$ into a matching by first scaling all vertex masses and transported masses by a factor of $n$ and then ensuring that the plan is integral using the cycle canceling method introduced by Kang and Payor (2015) in near-linear time. This produces a perfect matching $M'$ of points in $A$ and $B$. We then remove all edges from the matching with $\mathrm{d}(a, b) = 1$ (those not originally present in $G$), which removes at most $\delta^p \cdot n$ edges (i.e., $|M'| \geq |M^*| - \delta^p \cdot n$), as all removed edges have a cost of 1. We then match the remaining $\delta^p \cdot n$ unmatched vertices using an augmenting path per unmatched vertex, with each path being found in $O(n^2)$ time. Thus, the final running time for the reduction is the sum of the time for our algorithm $\mathbb{A}$ and the time for remaining augmentations. Explicitly, we have a running time of $T(n, \delta, p) + n^3 \cdot \delta^p$.

For any additive approximate solver with $T(n, \delta, p) = n^2/\delta^{(1-\epsilon)p}$, we can select $\delta = 1/n^{\frac{1+\epsilon}{2p}}$ and obtain an $n^{2.5-(\epsilon^2/2)}$ time algorithm for finding a perfect matching. Our reduction shows that it is unlikely that the existing techniques can be adapted to obtain a $\delta$-additive approximation algorithm for $p$-Wasserstein distance with an execution time of $O(n^2/\delta^{p(1-\varepsilon)})$ for any constant $\varepsilon > 0$.

## 4.2. Algorithm for RPW distance

In this section, we present an additive approximation algorithm for the $(p,k)$-RPW distance, for any $p \geq 1$. For simplicity in presentation, we describe our algorithm assuming $k = 1$ and refer to the distance as the $p$-RPW distance, noting that a similar approach easily extends to any constant $k > 0$. For two distributions $\mu$ and $\nu$, let $w^*$ be the $p$-RPW distance between them, i.e., $w^*$ is the smallest value such that $W_{p,1-w^*}(\mu, \nu) \leq w^*$. Any value $w$ is $\delta$-close to the $p$-RPW distance if $|w - w^*| \leq \delta$.

At a high level, given any guess value $g$ of $w^*$, our algorithm computes a partial transport plan $\sigma_g$ and uses the cost and the amount of transported mass in $\sigma_g$ to obtain an estimate $w_g$, which is an upper bound (up to an additive error $\delta$) on the $p$-RPW distance. We show that, interestingly, when $g \in [w^*, w^* + \frac{\delta}{4})$, the computed transport plan $\sigma_g$ transports at least $1 - w^* - \delta$ mass with a cost at most $w^* + \delta$. We use this observation to show that the returned estimate $w_g$ would be a $\delta$-close $p$-RPW distance. Consider the series $g_i = \frac{i\delta}{4}$ for $i = 1, \ldots, \frac{4}{\delta}$. Note that at least one of the $g_i$ values is at most $\frac{\delta}{4}$ away from $w^*$. We execute our algorithm for all $g_i$'s and return the smallest upper bound as a $\delta$-close $p$-RPW distance.

Our algorithm executes a modified version of the LMR algorithm (Lahn et al., 2019). Given a parameter $\varepsilon > 0$, the LMR algorithm computes, for each value $\alpha \in [0,1]$, an $\varepsilon$-close $\alpha$-partial transport plan by first scaling the problem instance to integer supplies, demands, and costs, and then executing a number of phases of Gabow and Tarjan's integer transportation algorithm (Gabow and Tarjan, 1989) (from now on referred to as the GT algorithm). We present an algorithm that, in a similar fashion, first transforms the problem instance to a scaled space, where all supplies, demands, and costs are integral, then executes several phases of the GT algorithm with a modified stopping condition, and finally projects the transport plan back to the original problem space. Next, we describe our algorithm in detail.

Given a guess value $g$, our algorithm lifts the problem instance to a scaled space as follows: Let $s_1 = \frac{2pn}{\delta}$ and $s_2 = \frac{g^p \delta}{2p}$. For any point $a \in A$ (resp. $b \in B$), define the scaled mass $\hat{\mu}(a) := \lceil s_1 \cdot \mu(a) \rceil$ (resp. $\hat{\nu}(b) := \lfloor s_1 \cdot \nu(b) \rfloor$). For any pair of points $(a,b) \in A \times B$, we set their distance to be $\hat{d}(a,b) = \left\lceil \frac{d(a,b)^p}{s_2} \right\rceil$. For any transport plan $\hat{\sigma}$ in the lifted space, let $\hat{w}(\hat{\sigma})$ denote the 1-Wasserstein cost of $\hat{\sigma}$ under $\hat{d}(\cdot, \cdot)$ distances.

At any point in the algorithm, we can also project our current transport plan back to our original problem space as follows: given a lifted transport plan $\hat{\sigma}$, we project it to the original problem space by defining a transport plan $\sigma_g$, where $\sigma_g(a,b) = \hat{\sigma}(a,b)/s_1$ for all pairs $(a,b) \in A \times B$.

Note that the rounding of masses might create extra mass on the points of $A$. Thus, when projecting back to the original space, our algorithm removes such excess mass as described in (Lahn et al., 2019).

Our algorithm initializes $\hat{\sigma}$ to an empty transport plan and executes $j^* := \frac{8p}{\delta^2}$ phases of the GT algorithm on the scaled space, which builds a partial transport plan incrementally in each phase using a primal-dual Hungarian-based approach. In each phase, the GT algorithm computes a set of augmenting paths and augments $\hat{\sigma}$ along those paths. For each phase $1 \leq j \leq j^*$, let $\hat{\sigma}_j$ denote the partial transport plan $\hat{\sigma}$ maintained after the execution of the $j$th phase. Define $\hat{w}_j := \hat{w}(\hat{\sigma}_j)$, define $\alpha_j$ as the amount of mass transported by $\hat{\sigma}_j$, and let $\hat{m}_j = \max\left\{ 1 - \frac{\alpha_j}{s_1}, (\frac{\hat{w}_j \cdot s_2}{s_1})^{1/p} \right\}$; here, the first term is an estimate of the untransported mass when we project $\hat{\sigma}_j$ back to the original space, and the second term is an estimate of projected cost. Note that as our algorithm executes the phases (i.e., $j$ increases), the value $\hat{m}_j$ will decrease until the estimated projected cost surpasses the estimated projected untransported mass.

If $(\frac{\hat{w}_j \cdot s_2}{s_1})^{1/p} \leq 1 - \frac{\alpha_j}{s_1}$ for all phases $j \leq j^*$, then our algorithm projects the transport plan $\hat{\sigma}_{j^*}$ to a transport plan $\sigma_g$ on the original space and returns $\hat{m}_g := \hat{m}_{j^*}$ as an estimate of the $p$-RPW distance.

Otherwise, suppose at an intermediate phase $j \leq j^*$, the cost $(\frac{\hat{w}_j \cdot s_2}{s_1})^{1/p}$ exceeds the mass $1 - \frac{\alpha_j}{s_1}$ for the first time, i.e., $(\frac{\hat{w}_j \cdot s_2}{s_1})^{1/p} \geq 1 - \frac{\alpha_j}{s_1}$, and, for all $k < j$, $(\frac{\hat{w}_k \cdot s_2}{s_1})^{1/p} < 1 - \frac{\alpha_k}{s_1}$. For any $\alpha \in (\alpha_{j-1}, \alpha_j]$, let $\hat{\sigma}^\alpha$ denote the $\alpha$-partial transport plan computed during the execution of the $j$th phase of the GT algorithm, i.e., the transport plan obtained after augmenting $\hat{\sigma}_{j-1}$ by $\alpha - \alpha_{j-1}$ mass. Let $\alpha^*$ be the value in $(\alpha_{j-1}, \alpha_j]$ such that $(\frac{\hat{w}(\hat{\sigma}^{\alpha^*}) \cdot s_2}{s_1})^{1/p} = 1 - \frac{\alpha^*}{s_1}$. As discussed by Phatak et al. (2022) and Raghvendra et al. (2024), the value $\alpha^*$ can be computed by taking the intersection point of a line segment and with the curve $y = (1 - \frac{x}{s_1})^p$ in constant time. Our algorithm then projects the transport plan $\hat{\sigma}^{\alpha^*}$ to a transport plan $\sigma_g$ in the original space and returns $\sigma_g$ as well as the value $\hat{m}_g = \max\left\{ 1 - \frac{\alpha^*}{s_1}, (\frac{\hat{w}(\hat{\sigma}^{\alpha^*}) \cdot s_2}{s_1})^{1/p} \right\} = 1 - \frac{\alpha^*}{s_1}$ as an estimate of the $p$-RPW distance given the guess $g$. This completes the description of our algorithm.

**Efficiency:** For each guess value $g$, our algorithm executes at most $j^* = \frac{8p}{\delta^2}$ phases of the GT algorithm, each in $O(n^2)$ time. Projecting the input instance to the lifted space and converting the computed transport plan back to the original space also takes $O(n^2)$ time. Consequently, our algorithm processes each guess value $g$ in $O(\frac{n^2}{\delta^2})$ time; summing across all $O(1/\delta)$ guess values, the running time of our algorithm would be $O(\frac{n^2}{\delta^3})$.

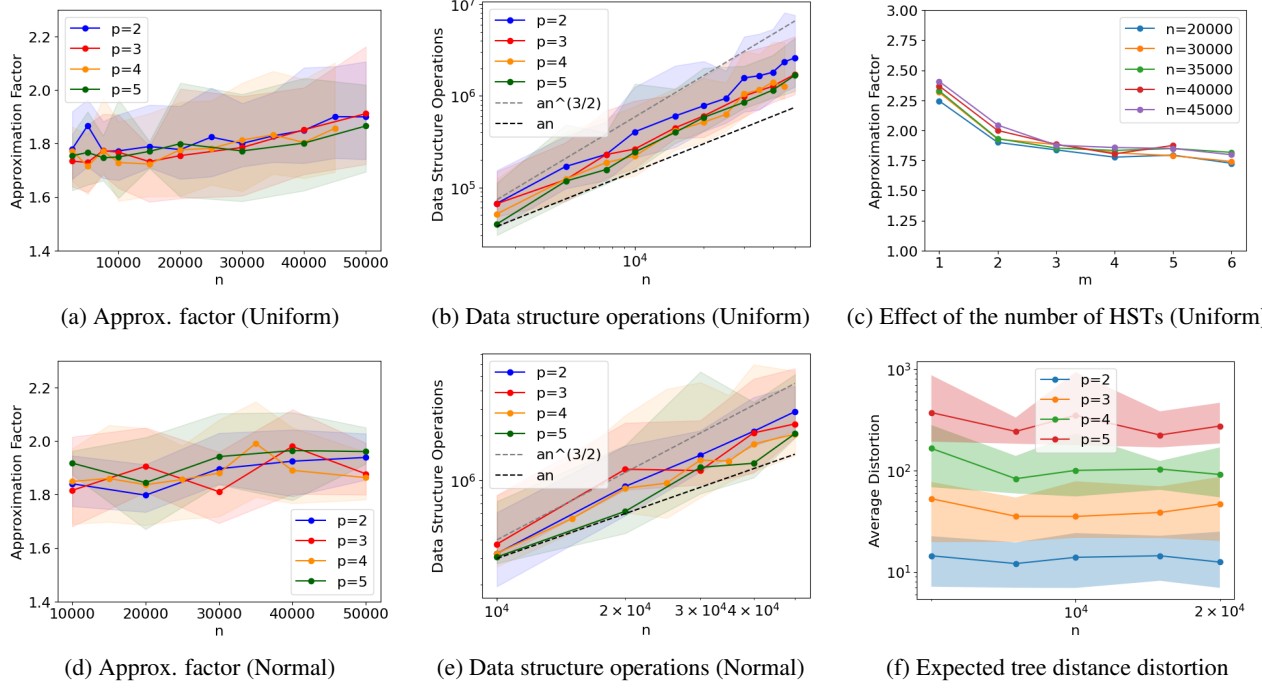

Figure 1. Our experimental results, (a) and (d): the approximation factor of our computed matching as a function of $n$, (b), (e): the number of calls to the data structure operations by the Hungarian algorithm, in log-log scale, (c): the approximation factor of our computed matching for $p = 4$ as varying the number of HSTs, and (f) the average distortion of the tree distances as a function of $n$, in log-log scale.

**Correctness:** For each guess $g_i$, we show that our algorithm returns a value that is guaranteed to be at least $w^* - \delta/16$; here $w^*$ is the optimal $(p, 1)$-RPW distance. Furthermore, we show that for any guess $g^* \in [w^*, w^* + \delta/4)$, our algorithm returns a value that is at most $w^* + \delta$. See Appendix B for a detailed analysis.

## 5. Experiments

In this section, we present the empirical analysis of both our tree-based distance from Section 2.2 as well as the algorithm from (Section 3). All computations are performed on a computer with a 2.6 GHz 6-Core Intel Core i7 CPU and 16 GB RAM, using a single calculation thread. The code is available at https://github.com/pouyansh/Faster-pWasserstein.

**Datasets:** We perform experiments on two datasets, namely (i) samples drawn from the uniform distribution inside the 10-dimensional unit hypercube (Uniform dataset), and (ii) samples drawn from a truncated normal distribution inside the 10-dimensional unit hypercube (Normal dataset).

**Results:** We present our results in three parts:

*Tree Distances Accuracy:* For two sets of $n$ points from the Uniform dataset, we compute the average distortion of the tree distance across all pairs of points. As depicted in Figure 1(f), the average distortion does not significantly

change as a function of $n$, while it increases exponentially as $p$ increases, which is in line with our result in Lemma 2.1.

*Algorithm's Accuracy:* For a given value of $n$ and $p$ and both datasets, we executed our algorithm on $n$ samples $A$ and $B$ from the dataset and computed the ratio of our matching cost to the optimal matching cost. For each value of $n$ and $p$, we report the average ratio over 15 executions. As shown in Figure 1(a) and (d), as $n$ grows, the approximation factors slightly increase; however, the approximation factors do not change as a function of $p$. We also measured the accuracy of our algorithm by varying the number of HSTs used in our construction. For $p = 4$, we varied the number of HSTs $m$ from 1 to 6 and measured the approximation ratio of the computed matching. As shown in Figure 1(c), introducing more than 4 HSTs seems to have an insignificant effect on the algorithm's accuracy.

*Algorithm's Efficiency:* We analyze the complexity of our algorithm by counting the number of calls made to the data structure. As we can see in Figure 1(b) and (e), our experiments suggest that the number of data structure operations grows faster than linear but sub-$n^{1.5}$, which is significantly better than the worst-case upper bound of $O(n^2)$. Interestingly, this empirical behavior is reminiscent of the performance guarantees established for the scaling algorithm of (Sharathkumar and Agarwal, 2012a).

## Acknowledgment

This research was supported by NSF grants CCF 2514753 and CCF 2223871. We would also like to acknowledge Advanced Research Computing (ARC) at Virginia Tech as well as A Root Cluster (ARC) at North Carolina State University, which provided us with the computational resources used to run the experiments.

## Impact Statement

This paper presents work whose goal is to advance the field of Machine Learning. There are many potential societal consequences of our work, none which we feel must be specifically highlighted here.

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

## A. Missing Proofs and Details of Section 2

**Lemma 2.2.** $\mathbb{E}[w_{\mathcal{T},p}(\sigma_{\mathcal{T}}^*)] = O((\log n)w_p(\sigma^*))$.

*Proof.* From the definition of the $p$-Wasserstein distance and since $\sigma_{\mathcal{T}}^*$ is an optimal transport plan with respect to the tree distances,

$$
\mathbb{E}\left[w_{\mathcal{T},p}(\sigma_{\mathcal{T}}^*)\right] = \mathbb{E}\left[\left(\sum_{(a,b)\in A\times B} \sigma_{\mathcal{T}}^*(a,b)\mathrm{d}_{\mathcal{T}}(a,b)^p\right)^{1/p}\right] \le \mathbb{E}\left[\left(\sum_{(a,b)\in A\times B} \sigma^*(a,b)\mathrm{d}_{\mathcal{T}}(a,b)^p\right)^{1/p}\right]
$$

$$
\le \left(\mathbb{E}\left[\sum_{(a,b)\in A\times B} \sigma^*(a,b)\mathrm{d}_{\mathcal{T}}(a,b)^p\right]\right)^{1/p}, \tag{5}
$$

where the last inequality holds by Jensen's inequality (Dekking et al., 2006). Using the linearity of expectation and Lemma 2.1, from Equation (5) and for some constants $c_1, c_2 > 0$,

$$
\mathbb{E}\left[w_{\mathcal{T},p}(\sigma_{\mathcal{T}}^*)\right] \le \left(\mathbb{E}\left[\sum_{(a,b)\in A\times B} \sigma^*(a,b)\mathrm{d}_{\mathcal{T}}(a,b)^p\right]\right)^{1/p} = \left(\sum_{(a,b)\in A\times B} \sigma^*(a,b)\,\mathbb{E}\left[\mathrm{d}_{\mathcal{T}}(a,b)^p\right]\right)^{1/p}
$$

$$
\le \left(\sum_{(a,b)\in A\times B} \sigma^*(a,b) \times c_1(4\log n)^p \mathrm{d}(a,b)^p\right)^{1/p} = (c_2\log n)\left(\sum_{(a,b)\in A\times B} \sigma^*(a,b) \times \mathrm{d}(a,b)^p\right)^{1/p}
$$

$$
= (c_2\log n)w_p(\sigma^*).
$$

$\square$

**Lemma 2.3.** *Suppose $(a^*,b^*) \in V \times U$ is the weighted bichromatic closest pair between $V$ and $U$. Then,*

$$
\mathrm{d}_{\mathcal{T},\mathrm{w}}(a^*,b^*)
$$
$$
= \min_{T_i\in\mathcal{T},C\in T_i}\left\{2^{-\mathrm{lev}(C)\cdot p} - \max_{b\in U_C}\mathrm{w}(b) - \max_{a\in V_C}\mathrm{w}(a)\right\}.
$$

*Proof.* For any cluster $C$ at a level $j$ of a tree $T_i \in \mathcal{T}$, define $\mathrm{d}(C) := 2^{-\mathrm{lev}(C)} = 2^{-j}$. Define the *weighted $C$-distance* of $a$ and $b$ as $\mathrm{d}_{C,\mathrm{w}}(a,b) := \mathrm{d}(C)^p - \mathrm{w}(a) - \mathrm{w}(b)$ and the *weighted $C$-closest pair* as the pair $(a^*,b^*) \in V_C \times U_C$ minimizing the weighted $C$-distance. Consider any tree $T_i \in \mathcal{T}$, any cluster $C \in T_i$ at a level $j$, and any pair $(a,b) \in V_C \times U_C$. From the definition of the tree distances,

$$
\mathrm{d}_{\mathcal{T},\mathrm{w}}(a,b) = \mathrm{d}_{\mathcal{T}}(a,b)^p - \mathrm{w}(a) - \mathrm{w}(b) \le \left(2^{-\mathrm{lev}_i(a,b)}\right)^p - \mathrm{w}(a) - \mathrm{w}(b) \le \mathrm{d}(C)^p - \mathrm{w}(a) - \mathrm{w}(b).
$$

Furthermore, if $C$ is the cluster determining the tree distance of $a$ and $b$ (i.e., the tree distance of $a$ and $b$ is determined by the tree $T_i$ and $C$ is the least common ancestor of $a$ and $b$ in $T_i$), then

$$
\mathrm{d}_{\mathcal{T},\mathrm{w}}(a,b) = \mathrm{d}(C)^p - \mathrm{w}(a) - \mathrm{w}(b).
$$

Therefore, for any pair of points $(a,b) \in V \times U$, if $\mathcal{C}_{a,b}^i$ denotes the set of all clusters of $T_i$ containing both $a$ and $b$,

$$
\mathrm{d}_{\mathcal{T},\mathrm{w}}(a,b) = \min_{i\in[1,p],C\in\mathcal{C}_{a,b}^i}\{\mathrm{d}(C)^p - \mathrm{w}(a) - \mathrm{w}(b)\}. \tag{6}
$$

Consequently, if $(a^*,b^*) \in V \times U$ is the weighted bichromatic closest pair, then

$$
\mathrm{d}_{\mathcal{T},\mathrm{w}}(a^*,b^*) = \min_{(a,b)\in V\times U}\mathrm{d}_{\mathcal{T},\mathrm{w}}(a,b) = \min_{(a,b)\in V\times U,\,T_i\in\mathcal{T},\,C\in\mathcal{C}_{a,b}^i}\mathrm{d}_{C,\mathrm{w}}(a,b)
$$

$$
= \min_{T_i\in\mathcal{T},\,C\in T_i,\,(a,b)\in V_C\times U_C}\mathrm{d}_{C,\mathrm{w}}(a,b) = \min_{T_i\in\mathcal{T},\,C\in T_i}\left\{\min_{(a,b)\in V_C\times U_C}\mathrm{d}_{C,\mathrm{w}}(a,b)\right\}. \tag{7}
$$

In other words, to find the weighted bichromatic closest pair, one can compute the weighted closest pair inside each cluster in all trees in $\mathcal{T}$ and find the closest one among such pairs. Finally, from Equation (6),

$$\min_{(a,b)\in V_C\times U_C} \mathrm{d}_{C,\mathrm{w}}(a,b) = \min_{(a,b)\in V_C\times U_C}\{\mathrm{d}(C)^p - \mathrm{w}(a) - \mathrm{w}(b)\} = \mathrm{d}(C)^p - \max_{a\in V_C}\mathrm{w}(a) - \max_{b\in U_C}\mathrm{w}(b). \tag{8}$$

Therefore, the weighted bichromatic closest pair for the cluster $C$ would be the pair $(a,b)\in V_C\times U_C$ where $a$ (resp. $b$) has the largest dual weight among all points in $V_C$ (resp. $U_C$). Combining Equations (7) and (8),

$$\mathrm{d}_{\mathcal{T},\mathrm{w}}(a^*,b^*) = \min_{T_i\in\mathcal{T},\, C\in T_i}\left\{\min_{(a,b)\in V_C\times U_C}\mathrm{d}_{C,\mathrm{w}}(a,b)\right\} = \min_{T_i\in\mathcal{T},C\in T_i}\left\{\mathrm{d}(C)^p - \max_{a\in V_C}\mathrm{w}(a) - \max_{b\in U_C}\mathrm{w}(b)\right\},$$

as claimed. $\qquad\square$

## B. Missing Proofs and Details of Section 4.2

In this section, we show that the value $w$ returned by our algorithm from Section 4.2 is a $\delta$-additive approximation of the $p$-RPW distance. Recall that our algorithm computed an estimate $\hat{\mathbf{m}}_{g_i}$ for any guess value $g_i = \frac{i\delta}{4}$ for all $i\in[1,\delta/4]$. In our proof, we first show that for any guess value $g$, the computed estimate $\hat{\mathbf{m}}_g$ is at least $w^* - \frac{\delta}{2p}$, where $w^*$ is the $p$-RPW distance (Lemma B.1). We then show that for a guess value $g^*\in[w^*,w^*+\delta/4)$, the computed estimate $\hat{\mathbf{m}}_{g^*}$ is at most $w^* + \delta$ (Lemma B.3). Combining the two lemmas, by picking the minimum value among all values in $\{\hat{\mathbf{m}}_1,\ldots,\hat{\mathbf{m}}_{4/\delta}\}$, our algorithm correctly returns a $\delta$-additive approximation of the $p$-RPW distance.

**Lemma B.1.** *For any value $g\in(0,1)$, let $\hat{\mathbf{m}}_g$ denote the value returned by our algorithm given the guess value $g$. Then, $\hat{\mathbf{m}}_g \geq w^* - \frac{\delta}{2p}$.*

**Proof of Lemma B.1.** We begin by bounding the change in the cost and the transported mass of a transport plan $\hat{\sigma}$ when projecting to the original space.

**Lemma B.2.** *For any value $g\in(0,1)$, let $\hat{\sigma}$ denote an $\alpha$-partial transport plan in the scaled space, and let $\sigma$ denote its projection to the original space. Then, $\sigma$ transports at least $\frac{\alpha}{s_1} - \frac{\delta}{2p}$ mass and has a cost $w_p(\sigma) \leq \left(\frac{\hat{w}(\hat{\sigma})s_2}{s_1}\right)^{1/p}$.*

*Proof.* Recall that our algorithm constructs the scaled space by defining $\hat{\mu}(a) = \lceil s_1\cdot\mu(a)\rceil$ (resp. $\hat{\nu}(b) = \lfloor s_1\cdot\nu(b)\rfloor$) for each point $a\in A$ (resp. $b\in B$), where $s_1 = \frac{2pn}{\delta}$. Therefore, for any point $a\in A$, the transport plan $\hat{\sigma}$ transports at most $\hat{\mu}(a) \leq s_1\cdot\mu(a) + 1$ mass from $a$ to the points of $B$, i.e., for any point $a\in A$,

$$\sum_{b\in B}\frac{\hat{\sigma}(a,b)}{s_1} \leq \frac{\hat{\mu}(a)}{s_1} \leq \mu(a) + \frac{1}{s_1}, \tag{9}$$

where the first inequality holds since $\hat{\sigma}$ is a partial transport plan. Recall that our algorithm projects $\hat{\sigma}$ back to the original space by first defining a transport plan $\sigma$ transporting $\sigma(a,b) = \hat{\sigma}(a,b)/s_1$ mass between each pair $(a,b)\in A\times B$. From Equation (9), for any point $a\in A$, the transport plan $\sigma$ transports up to $\frac{1}{s_1} = \frac{\delta}{2pn}$ excess mass, which our algorithm then adjusts by arbitrarily removing masses on the edges incident on $a$. Therefore,

$$\sum_{a\in A}\sum_{b\in B}\sigma(a,b) \geq \sum_{a\in A}\left(\left(\sum_{b\in B}\frac{\hat{\sigma}(a,b)}{s_1}\right) - \frac{1}{s_1}\right) = \frac{\sum_{(a,b)\in A\times B}\hat{\sigma}(a,b)}{s_1} - \frac{n}{s_1} = \frac{\alpha}{s_1} - \frac{\delta}{2p}. \tag{10}$$

Next, we show that the cost of $\sigma$ is at most $\frac{\hat{w}(\hat{\sigma})\cdot s_2}{s_1}$. Recall that our algorithm defines the scaled distance of each pair $(a,b)\in A\times B$ as $\hat{\mathrm{d}}(a,b) = \left\lceil\frac{\mathrm{d}(a,b)^p}{s_2}\right\rceil$ for $s_2 = \frac{g^p\delta}{2p}$. Since $\frac{\mathrm{d}(a,b)^p}{s_2} \leq \hat{\mathrm{d}}(a,b)$ and $\sigma(a,b) \leq \hat{\sigma}(a,b)/s_1$,

$$(w_p(\sigma))^p = \sum_{(a,b)\in A\times B}\sigma(a,b)\cdot\mathrm{d}(a,b)^p \leq \sum_{(a,b)\in A\times B}\frac{\hat{\sigma}(a,b)}{s_1}\cdot s_2\hat{\mathrm{d}}(a,b) = \frac{\hat{w}(\hat{\sigma})\cdot s_2}{s_1}.$$

$\qquad\square$

For any transport plan $\sigma$ (in the original space), let $\alpha(\sigma)$ denote the amount of mass transported by $\sigma$. As shown in Lemma 2.2 in (Raghvendra et al., 2024), if $w^*$ denotes the $p$-RPW distance between $\mu$ and $\nu$, then $w^* \leq \max\{1 - \alpha(\sigma), w_p(\sigma)\}$. Given a guess value $g \in [0, 1]$, recall that our algorithm computes an $\hat{\alpha}$-partial transport plan $\hat{\sigma}$ in the scaled space and returns a value $\hat{\mathbf{m}}_g = \max\{1 - \frac{\hat{\alpha}}{s_1}, (\frac{\hat{w}(\hat{\sigma}) \cdot s_2}{s_1})^{1/p}\}$ and a projected transport plan $\sigma_g$. From Lemma B.2, $\alpha(\sigma_g) \geq \frac{\hat{\alpha}}{s_1} - \frac{\delta}{2p}$ and $w_p(\sigma_g) \leq (\frac{\hat{w}(\hat{\sigma}) \cdot s_2}{s_1})^{1/p}$. Hence,

$$\hat{\mathbf{m}}_g = \max\left\{1 - \frac{\hat{\alpha}}{s_1}, \left(\frac{\hat{w}(\hat{\sigma}) \cdot s_2}{s_1}\right)^{1/p}\right\} \geq \max\left\{1 - \alpha(\sigma_g) - \frac{\delta}{2p}, w_p(\sigma_g)\right\} \geq w^* - \frac{\delta}{2p}.$$

This completes the proof of Lemma B.1.

**Lemma B.3.** *For any value $g^*$ with $g^* \in [w^*, w^* + \frac{\delta}{4})$, suppose $\hat{\mathbf{m}}_{g^*}$ denotes the returned estimate of the $p$-RPW for the guess $g^*$. Then, $\hat{\mathbf{m}}_{g^*} \leq w^* + \delta$.*

**Proof of Lemma B.3.** Let $\hat{\sigma}_{g^*}$ denote the transport plan in the scaled space computed by our algorithm for the guess value $g^*$, and let $\sigma_{g^*}$ denote the projection of $\hat{\sigma}_{g^*}$ to the original space. To prove Lemma B.3, we first show in Lemma B.5 that the transport plan $\hat{\sigma}_{g^*}$ transports at least $s_1(1 - w^* - \delta)$ mass, i.e., if $\hat{\alpha}$ denotes the amount of mass transported by $\hat{\sigma}_{g^*}$, then

$$1 - \frac{\hat{\alpha}}{s_1} \leq w^* + \delta. \tag{11}$$

We then show in Lemma B.6 that in the scaled space,

$$\left(\frac{s_2 \hat{w}(\hat{\sigma}_{g^*})}{s_1}\right)^{1/p} \leq w^* + \delta. \tag{12}$$

Combining Equations (11) and (12), $\hat{\mathbf{m}}_{g^*} \leq w^* + \delta$, as claimed.

The following lemma helps in proving Lemmas B.5 and B.6.

**Lemma B.4.** *Let $\sigma^*$ denote an optimal $(1 - w^*)$-partial transport plan on the original space, and let $\hat{\sigma}^*$ denote its projection to the scaled space. Then, $\hat{w}(\hat{\sigma}^*) \leq \frac{s_1(w^*)^p}{s_2} + s_1$. Furthermore, the total amount of mass transported by $\hat{\sigma}^*$ is at least $s_1(1 - w^* - \frac{\delta}{2p})$.*

*Proof.* From the definition of the scaled distances as well as the projection of transport plans to the scaled space,

$$\hat{w}(\hat{\sigma}^*) = \sum_{(a,b) \in A \times B} \hat{\sigma}^*(a,b) \cdot \hat{\mathrm{d}}(a,b) \leq \sum_{(a,b) \in A \times B} s_1 \sigma^*(a,b) \cdot \left(\frac{\mathrm{d}(a,b)^p}{s_2} + 1\right) = \frac{s_1(w_p(\sigma^*))^p}{s_2} + s_1(1 - w^*)$$
$$\leq \frac{s_1(w^*)^p}{s_2} + s_1.$$

For any point $b \in B$, the transport plan $\sigma$ transports at most $\nu(b) \leq \frac{\hat{\nu}(b)+1}{s_1}$ mass from $b$ to the points of $A$, i.e., for any point $b \in B$,

$$\sum_{a \in A} s_1 \sigma(a,b) \leq s_1 \nu(b) \leq \hat{\nu}(b) + 1. \tag{13}$$

After defining a scaled transport plan $\hat{\sigma}$ transporting $\hat{\sigma}(a,b) = s_1 \sigma(a,b)$ mass between each pair $(a,b) \in A \times B$, from Equation (13), for any point $b \in B$, the transport plan $\hat{\sigma}$ transports up to 1 unit of excess mass, which our algorithm then adjusts by arbitrarily removing masses on the edges incident on $b$. Therefore,

$$\sum_{b \in B} \sum_{a \in A} \hat{\sigma}(a,b) \geq \sum_{b \in B} \left(\left(\sum_{a \in A} s_1 \sigma(a,b)\right) - 1\right) = s_1 \sum_{(a,b) \in A \times B} \hat{\sigma}(a,b) - n = s_1(1 - w^*) - n \geq s_1\left(1 - w^* - \frac{\delta}{2p}\right).$$

$\square$

**Lemma B.5.** *For the guess $g^*$ with $0 \le g^* - w^* \le \frac{\delta}{4}$, let $\hat{\alpha}_{j^*}$ denote the amount of mass transported by the transport plan $\hat{\sigma}_{j^*}$ at the end of phase $j^*$. Then, $1 - \frac{\hat{\alpha}_{j^*}}{s_1} \le w^* + \delta$.*

*Proof.* We begin by introducing the notations necessary to prove this lemma. For any augmenting path $P$, let the *net-cost* of $P$, denoted by $\phi(P)$, be the rate of change in the cost of the transport plan when augmented along $P$, i.e., augmenting the transport plan $\sigma$ along the path $P$ by $\beta$ mass would increase the cost of $\sigma$ by $\beta \times \phi(P)$. Each phase $j$ of Gabow and Tarjan's algorithm is associated with a net-cost $\phi_j$, and in phase $j$, it finds and augments the maintained transport plan along a set of augmenting paths, each with a net-cost $\phi_j$. It is shown that

(P1) in each phase $j$, the net-cost value $\phi_j$ increases by at least one, i.e., $\phi_j \ge \phi_{j-1} + 1$ (Gabow and Tarjan, 1991).

For a transport plan $\sigma$ in the original space, we project $\sigma$ to the scaled space as follows. We first set $\hat{\sigma}(a, b) = s_1 \sigma(a, b)$ for all pairs $(a, b) \in A \times B$, and, for each point $b \in B$ such that the amount of mass transported out of $b$ according to $\hat{\sigma}$ is more than $\hat{\nu}(b)$, we arbitrarily remove mass transportation on the edges incident to it.

Recall that our algorithm stops after executing $j^* = \frac{8p}{\delta^2}$ phases of the Gabow and Tarjan's algorithm. We show that the transport plan $\hat{\sigma}_{j^*}$ computed at the end of phase $j^*$ transports at least $s_1(1 - w^* - \delta)$ mass to conclude the lemma statement.

Suppose $\hat{\sigma}_{j^*}$ transports $s_1(1 - w^* - \frac{\delta}{2p}) - \beta$ mass, for some $\beta > 0$ (otherwise, the lemma statement holds trivially). Suppose, instead of stopping early, our algorithm would have continued executing the phases until transporting a total of $s_1(1 - w^* - \frac{\delta}{2p})$ mass, and let $\mathcal{P} = \langle P_1, P_2, \ldots, P_k \rangle$ denote the sequence of augmenting paths computed by the phases to transport the remaining $\beta$ mass. Note that by property (P1), all augmenting paths in $\mathcal{P}$ have net-costs more than $j^* = \frac{8p}{\delta^2}$. Therefore, if $\beta_i$ denotes the amount of mass augmented along each augmenting path $P_i$ and $\hat{\sigma}_{w^*}$ denotes the $s_1(1 - w^* - \frac{\delta}{2p})$-partial transport plan obtained after augmenting $\hat{\sigma}_{j^*}$ along the paths in $\mathcal{P}$, then

$$\hat{w}(\hat{\sigma}_{w^*}) - \hat{w}(\hat{\sigma}_{j^*}) = \sum_{i=1}^{k} \phi(P_i)\beta_i \ge \sum_{i=1}^{k} \frac{8p}{\delta^2}\beta_i = \frac{8p\beta}{\delta^2}. \tag{14}$$

Combined with Lemma B.4 and the property that each intermediate transport plan is $s_1$-close as shown in (Phatak et al., 2022),

$$\frac{8p\beta}{\delta^2} \le \hat{w}(\hat{\sigma}_{w^*}) - \hat{w}(\hat{\sigma}_{j^*}) \le \hat{w}(\hat{\sigma}_{w^*}) \le \hat{w}(\hat{\sigma}^*) + s_1 \le \frac{s_1(w^*)^p}{s_2} + 2s_1, \tag{15}$$

where the first inequality holds from Equation (14), the third inequality holds since $\hat{\sigma}_{j^*}$ is a $s_1$-close $s_1(1 - w^* - \frac{\delta}{2p})$-partial transport plan and $\hat{\sigma}^*$ is a transport plan that transports at least $s_1(1 - w^* - \frac{\delta}{2p})$ mass, and the last inequality holds from Lemma B.4. Recall that $g^* \ge w^*$. Plugging in the value $s_2 = \frac{\delta(g^*)^p}{2p}$ into Equation (15),

$$\beta \le \frac{s_1\delta^2}{8p}\left(\frac{2p(w^*)^p}{\delta(g^*)^p} + 2\right) \le s_1\delta \times \left(\frac{1}{4} + \frac{\delta}{4p}\right) \le s_1\delta \times \frac{1}{2}. \tag{16}$$

In other words, the transport plan $\hat{\sigma}_{j^*}$ computed at the end of phase $j^*$ transports at least $\alpha_{j^*} \ge s_1(1 - w^* - \frac{\delta}{2p}) - \frac{s_1\delta}{2}$ mass, and $1 - \frac{\alpha_{j^*}}{s_1} \le w^* + \delta$. $\qquad\square$

**Lemma B.6.** *For the guess $g^*$ with $0 \le g^* - w^* \le \frac{\delta}{4}$, any transport plan $\hat{\sigma}$ computed by our algorithm in the scaled space that transports a mass $\alpha \in [s_1(1 - w^* - \delta), s_1(1 - w^* - \frac{\delta}{2p})]$ satisfies $\left(\frac{s_2\hat{w}(\hat{\sigma})}{s_1}\right)^{1/p} \le w^* + \delta$.*

*Proof.* Recall that, by the properties of Gabow and Tarjan's algorithm, the transport plan $\hat{\sigma}$ would be a $s_1$-close $\alpha$-partial transport plan. If $\hat{\sigma}^*_\alpha$ denotes the $\alpha$-partial OT plan in the scaled space, then using Lemma B.4,

$$\hat{w}(\hat{\sigma}) \le \hat{w}(\hat{\sigma}^*_\alpha) + s_1 \le \hat{w}(\hat{\sigma}^*) + s_1 \le \frac{s_1(w^*)^p}{s_2} + 2s_1.$$

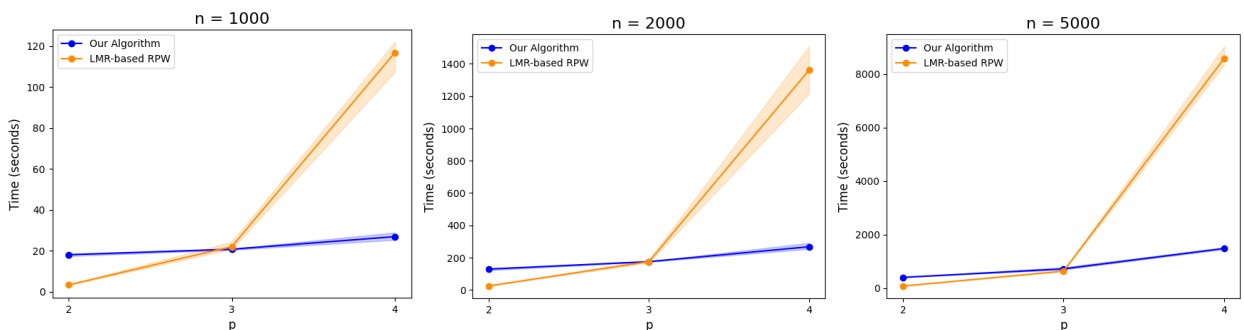

*Figure 2.* Comparing the running time of our RPW algorithm (blue lines) with the algorithm of (Raghvendra et al., 2024) (orange lines) as $p$ increases.

Therefore,

$$\frac{s_2}{s_1}\hat{w}(\hat{\sigma}) \leq \frac{s_2}{s_1}\left(\frac{s_1(w^*)^p}{s_2} + 2s_1\right) \leq (w^*)^p + 2s_2 = (w^*)^p + \frac{\delta(g^*)^p}{p} \leq (w^*)^p + \frac{\delta(w^* + \delta/4)^p}{p} \leq (w^* + \delta)^p,$$

where the last inequality holds since

$$(w^* + \delta)^p = (w^*)^p + \binom{p}{1}w^{*p-1}\delta + \binom{p}{2}w^{*p-2}\delta^2 + \ldots + \delta^p$$

$$\geq (w^*)^p + \left(\frac{w^*}{p} + \frac{p}{4p}\right)w^{*p-1}\delta + \frac{1}{p} \times \left(\binom{p}{2}w^{*p-2}\delta^2 + \ldots + \delta^p\right)$$

$$\geq (w^*)^p + \frac{1}{p} \times \left(w^{*p}\delta + \binom{p}{1}w^{*p-1}\frac{\delta}{4} + \ldots + \left(\frac{\delta}{4}\right)^p\right)$$

$$\geq (w^*)^p + \frac{\delta(w^* + \delta/4)^p}{p}.$$

$\square$

## C. Additional Experimental Results

In this section, we present an empirical comparison between our RPW algorithm presented in Section 4.2 and the current state-of-the-art RPW algorithm presented in (Raghvendra et al., 2024)(LMR-RPW). We demonstrate that the algorithm is both implementable and efficient. All experiments were performed on North Carolina State University's computing cluster (ARC), on a single calculation CPU thread. Implementation can be found at https://github.com/saarinenemma/FasterRPW.

**Results:** We perform experiments on samples drawn from the uniform distribution inside the unit square. For two sets of $n$ points, one sampled from the left half of the unit square and the second drawn from the right, we execute both LMR-RPW and our RPW algorithms to find the $p$-RPW distance between the datasets. We measure efficiency by comparing runtime in seconds. We present two strengths of our algorithm:

*Dependence on $p$:* For $\delta = 0.1$ and any $n \in \{1000, 2000, 5000\}$, we compute the $p$-RPW distance between $n$ samples from the left half and $n$ samples from the right half of the unit square for increasing values of $p$. As shown in Figure 2, for each sample size, the running time of our algorithm grows much slower as $p$ increases compared to LMR-RPW, and we observe that for $p = 4$, our algorithm computes an approximate $p$-RPW significantly faster than LMR-RPW.

*Dependence on $\delta$:* For $n = 1000$ and $p = 3$, we compute the $p$-RPW distance between two sets of $n$ samples drawn from each half of the unit square for $\delta \in \{0.1, 0.08, 0.06, 0.04\}$. As shown in Figure 3, as the additive error $\delta$ decreases (i.e., the accuracy increases), the runtime of the LMR-RPW algorithm increases at a greater rate, and our algorithm outperforms the LMR-RPW for all values of $\delta$.

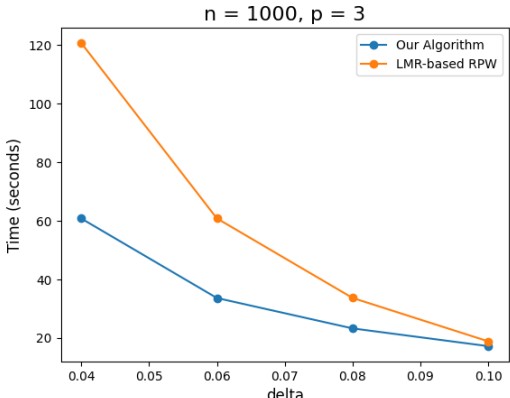

*Figure 3.* Comparing the running time of our RPW algorithm (blue lines) with the algorithm of (Raghvendra et al., 2024) (orange lines) as $\delta$ increases.

