# OpenReview forum: "Scalable Approximation Algorithms for $p$-Wasserstein Distance and Its Variants"
_ICML.cc/2025/Conference — ICML 2025 poster_

### Official Review · Reviewer_ZSVF · 2025-03-08

**Overall Recommendation:** 2

**Summary:**

This work introduces a method to compute a $O(\log n)$-approximation of the p-Wasserstein distance in $O(n^2 \log n)$ time for $p\ge 2$. The method is based on the construction of Hierarchically well Separated Trees (HSTs), and the Hungarian algorithm with Bichromatic Closest Pairs. Furthermore, the authors provide a method to approximate a variant of the p-Wasserstein distance which is robust to noise, named $(p,k)$-RPW, which can be computed in $O(n^2/\delta^2)$ for an additive error of $\delta$. This is an improvement compared to other methods such as entropic regularized OT computed with Sinkhorn which are computed in $O(n^2/\delta^p)$.


## Update after rebuttal

I maintain my score as I believe that additional experiments (especially for RPW) and comparisons would strengthen this work.

**Claims And Evidence:**

The authors claim that they provide a method to compute a $O(\log n)$-approximation of the p-Wasserstein distance in $O(n^2 \log n)$ time for $p\ge 2$ (Theorem 1.1), and that they can approximate the $(p,k)$-RPW distance in $O(n^2/\delta^3)$ time for an additive error of $\delta$ (Theorem 1.2). Both claims are supported by proofs. More precisely, in Section 2, a distance is derived, and is shown to be a $O(\log n)$ approximation of the Wasserstein distance, and in Section 3, an algorithm with a complexity of $O(n^2 \log n)$ to compute the proposed distance is described. Finally, the algorithm to approximate the RPW distance is derived in Section 4.

**Essential References Not Discussed:**

All the essential references seem to be discussed.

**Experimental Designs Or Analyses:**

The experimental design seems good. They demonstrate the complexity of the first algorithm in Figure 1.b and 1.e, and the approximation in Figure 1.f. They also verify the impact of several parameters on the results. However, there are no comparisons with baselines which are usually used to compute the $p$-Wasserstein distance. Moreover, a practical study of the algorithm to approximate RPW seems to be lacking.

**Methods And Evaluation Criteria:**

The methods proposed make sense. The evaluation criteria allow to verify the claims in practice. Nonetheless, I believe it would have been interesting to provide comparisons with known baselines, e.g. comparing the runtime of the method with the implementation of the OT problem in the library Python Optimal Transport. And likewise, it would have been nice to compare the RPW algorithm with e.g. Sinkhorn. Also, I am not sure the results of RPW are presented in Figure 1?

**Other Comments Or Suggestions:**

I suggest improving readibility by adding algorithms or figures to help the reader understand how to implement the algorithm.

It would also be nice to have experimental results for the computation of RPW, and comparisons with benchmark methods. For instance, the algorithm should be faster than the hungarian algorithm, but it does not appear on the experimental sections.

Maybe I missed it, but I am not sure what "$an$" and "$an^{(3/2)}$" refer to in Figure 1.b and 1.d.

Typos:
- Line 189, 1st column: "distace"

**Other Strengths And Weaknesses:**

**Strengths**
- Interesting method to compute an approximation of the $p$-Wasserstein distance in $O(n^2 \log n)$ time.
- Also provide an interesting method to compute an approximation of the $p$-RPW distance.
- Verify the theoretical results in practice.


**Weaknesses**
- The procedures proposed can be a bit hard to understand, as they are described in the text. Algorithms/Figures could maybe help the readers to better understand the methods.
- The experimental verifications are done only with the procedure to approximate the $p$-Wasserstein distance (if I understand well). Moreover, there are no comparisons with baselines which are usually used to compute the $p$-Wasserstein distance.

**Questions For Authors:**

1. Is your approximation algorithm faster than ot.emd in the library POT?
2. Did you implement the algorithm to compute RPW?
3. How does your algorithm introduced to compute RPW differ from the ones in [1]?

[1] Sharath Raghvendra, Pouyan Shirzadian, and Kaiyi Zhang. A new robust partial p-Wasserstein-based metric for comparing distributions. In 41st International Conference on Machine Learning, 2024.

**Relation To Broader Scientific Literature:**

One of the key contribution is to use a set of $p$ Hierarchically well Separated Tree Metrics instead of one, which was done for $p=1$ in [1, 2]. Another key contribution is to show a method to provide an approximation of the $p$-RPW distance, introduced in [3].

[1] Moses S Charikar. Similarity estimation techniques from rounding algorithms. In Proceedings of the thiry-fourth annual ACM symposium on Theory of computing, pages 380–388, 2002.

[2] Jon Kleinberg and Eva Tardos. Approximation algorithms for classification problems with pairwise relationships: Metric labeling and markov random fields. Journal of the ACM (JACM), 49(5):616–639, 2002.

[3] Sharath Raghvendra, Pouyan Shirzadian, and Kaiyi Zhang. A new robust partial p-Wasserstein-based metric for comparing distributions. In 41st International Conference on Machine Learning, 2024.

**Theoretical Claims:**

The theoretical claims seem good. The claims (see Claims and Evidence section) are supported through proofs in Section 2, 3, 4 and in Appendix. However, I did not check all the details.

---

> ### Author Rebuttal · Authors · 2025-03-31
>
> We thank the reviewer for their thorough review and constructive feedback. The two main contributions of our paper are as follows:
>
>
> * A $O(\log n)$ relative approximation algorithm for $p=1$ has been known for almost three decades and has had a significant impact. Despite significant effort, no such algorithm was known for $p=2$. We present the first $O(\log n)$ relative approximation algorithm for the $p$-Wasserstein distance, for any fixed value $p > 1$.
>
> * Under reasonable assumptions, we show that any algorithm that additively approximates $p$-Wasserstein distance requires $\Omega(n^2/\delta^p)$ time. This hardness also extends to other robust variants including $\lambda$-ROBOT and partial $p$-Wasserstein distance. In contrast, the $p$-RPW distance, due to its robustness, admits an $O(n^2/\delta^3)$ time algorithm for any value $p \ge 1$.
>
> We note that our contributions are significant from a theoretical standpoint. The primary reason to implement and test our relative approximation algorithm is to understand the gap between its worst-case theoretical guarantees and its practical performance. Having said that, we can include a comparison of our algorithm with the standard Hungarian algorithm as well as an implementation of our approximation algorithm of $p$-RPW with that of [1].
>
> **Comparison with OT.emd:** There are two reasons for not making a direct comparison between our algorithm and OT.emd.
>
> * OT.emd requires $O(n^2)$ space which means we can only execute them on small instances. This restricts our ability to compare our algorithm with OT.emd and Sinkhorn on larger inputs and understand how their performance scales with input size.
>
> * Our prototype implementation is written in Python where as OT.emd is based on a highly optimized C++ implementation. Comparing the performance of codes written in different languages, especially only in small instances, can mislead us which is why we have avoided a direct comparison.
>
> ----
> ----
>
> >Maybe I missed it, but I am not sure what "$an$" and "$an^{(3/2)}$" refer to in Figure 1.b and 1.d.
>
> **Response:** The plot uses a $\log$–$\log$ scale, where any polynomial function appears as a straight line, with the slope indicating the polynomial’s exponent. In Plot 1.b, we included the functions $f(n) = an$ and $f(n) = an^{3/2}$ (for some constant $a$) to illustrate that, in our experiments, the running time of the Hungarian algorithm with our data structure is bounded by $O(n^{3/2})$.
>
> ----
> ----
>
> >Did you implement the algorithm to compute RPW?
>
> **Response:** We show that additive approximation requires $\Omega(n^2/\delta^p)$ time due to their sensitivity to noise and the robustness of $p$-RPW allows for significantly faster additive approximation with an execution time of $O(n^2/\delta^3)$ (for all values of $p$). Our intention was to highlight this gap, which can be accomplished without the need for an implementation.
> Nonetheless, we are in the process of implementing our algorithm and commit to comparing it with the algorithm of [1] in the next version of our paper.
>
> ----
> ----
>
> >How does your algorithm introduced to compute RPW differ from the ones in [1]?
>
> **Response:**
> The algorithm introduced in [1] computes the OT profile and uses it to approximate the $p$-RPW, which takes $O(n^2/\delta^p+ n/\delta^{2p})$ time. We use a guessing procedure along with an early stopping criteria to reduce the execution time to $O(n^2/\delta^3)$. More precisely, our algorithm picks a guess $g$ from one of the $O(1/\delta)$ values as an approximation of the true $p$-RPW distance. For this value, it executes only $O(p/\delta^2)$ iterations of the LMR algorithm [2]. It then returns the minimum estimate achieved across all the $O(1/\delta)$ guesses. In contrast, the algorithm in [1] runs the LMR algorithm [2] for $O(1/\delta^p)$ iterations.
>
>
> [1] S. Raghvendra, P. Shirzadian, and K. Zhang. "A New Robust Partial $p$-Wasserstein-Based Metric for Comparing Distributions." ICML 2024.
>
> [2] N. Lahn, D. Mulchandani, and S. Raghvendra. "A graph theoretic additive approximation of optimal transport." NeurIPS 2019.

---

> > ### Comment · Reviewer_ZSVF · 2025-04-05
> >
> > Thank you for your detailed rebuttal and for clarifying the main contributions.
> >
> > I understand that it is mostly a theoretical work. The comparison with ot.emd might indeed be hard to compete with. A solution might be to reimplement it in python? For RPW, I still believe that a comparison of the implementations would really strengthen this work, even though the theoretical result is valuable on its own as underlined by the authors.

---

> > > ### Author Response · Authors · 2025-04-06
> > >
> > > Our work is primarily theoretical, and we have rigorously demonstrated its efficiency and correctness through detailed proofs. To ensure completeness, we will include additional comparisons that you have suggested. Specifically, we will compare the efficiency of our relative approximation algorithm against an exact solver, i.e., the Hungarian algorithm, and compare our RPW approximation with the algorithm proposed in [1].
> > >
> > > We sincerely thank the reviewer for highlighting the exact solver OT.emd and its efficiency. OT.emd is a C++ implementation of the network simplex algorithm. This implementation is detailed in [2]. A valuable direction for future research would involve comparing the trade-off between accuracy and efficiency across various C++ implementations of OT approximations. However, this analysis is beyond the scope of our current paper.
> > >
> > > [1] S. Raghvendra, P. Shirzadian, and K. Zhang. "A New Robust Partial $p$-Wasserstein-Based Metric for Comparing Distributions." ICML 2024.
> > >
> > > [2] N. Bonneel, M. Van De Panne, S. Paris, and W. Heidrich. "Displacement interpolation using Lagrangian mass transport". In ACM Transactions on Graphics (TOG), 2011.

---

### Official Review · Reviewer_GvkL · 2025-03-12

**Overall Recommendation:** 4

**Summary:**

This paper aims to provide an $O(\log n)$ approximation algorithm for p-Wasserstein distance that runs in $O(n^2 \log n \log U \log \Delta)$ time. This is done with a collection of $p$ HST trees and with a dynamic BCP data structure to efficiently find augmenting paths in the dual framework for the OT problem. The authors also present an algorithm for an additive approximation of a noise-resistant variant of $p$-Wasserstein called $(p,k)-RPW$. Their approach to this borrows ideas from the work of [Lahn et al. 19], scaling the problem to an integer one and using the [Gabow and Tarjan 89] algorithm. Some simple experiments demonstrate the scaling behavior that is expected, with also evidence that the average-case performance for the first method is between $n$ and $n^{3/2}$.

## Update after rebuttal:

I felt sufficiently satisfied with the explanations for my concerns, so I've updated my score to accept.

**Claims And Evidence:**

This is mostly a theoretical paper, so its evidence is mostly in the proofs. As noted in the section below, I was able to check Section 2 and Appendix A carefully, which included:
* a O(log n) in expectation approximation of the true p-Wasserstein distance with p HST trees
* a description of a data structure that allows for efficient construction, retrieval, and insertion/deletion
The proofs were correct. I was not able to check Section 3 (which used these tools to calculate their tree approximation distance in near-quadratic time) and Section 4/Appendix B (which proposed their method for additive approximation of $(p,k)$-RPW) in as much detail, but the techniques used were sensible.

There was also some empirical support for the theoretically established complexities via some simple experiments.

**Essential References Not Discussed:**

See above.

**Experimental Designs Or Analyses:**

The experiments were relatively simple, but reasonable, in my opinion.

**Methods And Evaluation Criteria:**

Yes, the experiments were reasonably done.

**Other Comments Or Suggestions:**

Rating Explanation: I appreciated the work and the technical depth of the approach to an important problem. My main reason for not providing a higher score is that the presentation seems a bit disjointed, with two related approximation problems solved with completely different techniques. Much of the intro describes noise sensitivity as an issue, but then the majority of text is spent on the first problem which is not looking at the $p-RPW$ distance.

One clarity nit: At the start of 2.1, I would remind people that one is considering a discrete, finite metric space. I found myself confused at $k^u_j$ being an integer for a moment.

**Other Strengths And Weaknesses:**

See below.

**Questions For Authors:**

1. Is the *spread* equal to the ratio of largest to smallest edge costs? These are both denoted with the same $\Delta$ notation, but are never explicitly connected.
2. In line 207, lev(a,b) is never explicitly defined, I don't think. It is easy enough to infer, but please clarify.
3. I don't think I understood the second inequality in the derivation 181-189 (second column, for inequality 1). It was my impression that the telescoping sum would eliminate $H_{k^a_0}$ and $H_{k^b_0}$ if $h \geq 2$. I believe the result still holds, but would have expected different terms. Please clarify.
4. In appendix A, for the proof of lemma 2.2, I expected the second equality to be a $\leq$, as the optimal transport plan for the base metric will be suboptimal for the tree metric, no?

**Relation To Broader Scientific Literature:**

The work does a good job of referencing and discussing prior work. I did not notice any glaring omissions.

One note though, is that I would have preferred that the authors mention the poor rate of convergence for empirical measures to underlying continuous measures in $p$-Wasserstein distance (a reference is [Weed & Bach 19]: *Sharp asymptotic ... empirical measures in Wasserstein distance*). This is a major practical limitation of the Wasserstein distance that felt conspicuously missing from the sentence in lines 054-055. I understand that it doesn't fit the story well, as I don't think this approach addresses this problem at all, but it felt wrong to omit it entirely.

**Theoretical Claims:**

I checked the arguments of Section 2 (including appendix A) in detail and am convinced of their correctness. Some questions on minor details in latter sections. For sections 3 and 4, I did not have time to check them carefully, but the parts in the main text seem fine upon a more cursory inspection.

---

> ### Author Rebuttal · Authors · 2025-03-31
>
> We thank the reviewer for their thorough review and constructive feedback. We will add a short discussion and a reference about the convergence rate of the Wasserstein distance, as well as the formal definition of spread and the corrected inequality in Appendix A in our next version. We answer the main concern raised by the reviewer below.
>
>
> >My main reason for not providing a higher score is that the presentation seems a bit disjointed, with two related approximation problems solved with completely different techniques. Much of the intro describes noise sensitivity as an issue, but then the majority of text is spent on the first problem.
>
>
> **Response:** Noise sensitivity of $p$-Wasserstein distance is the primary reason why both relative and additive approximations are challenging to design. We elaborate on this below and edit the paper to include this discussion.
>
> *Relative Approximations:* Embedding a metric into a tree metric introduces noise, or distortion, to the edge costs. Due to the high sensitivity, the optimal $p$-Wasserstein distance with respect to these noisy edge costs fails to serve as a relative approximation to the $p$-Wasserstein distance with respect to the original costs, i.e., the approximation quality is unbounded. To address this, we reduce the distortion in edge costs by selecting the smallest distortion across $p$ different tree embeddings. We demonstrate that the optimal $p$-Wasserstein distance computed with respect to these edge costs is in fact a $O(\log n)$-approximation of the true $p$-Wasserstein distance.
>
>
> *Additive Approximations:*  A small noisy mass of $\delta^p$ can affect the $p$-Wasserstein distance by $\delta$. Consequently, any $\delta$-additive approximation algorithm must transport all but $\delta^p$ mass in an approximately minimum-cost way to ensure a bound of $\delta$ on the additive approximation, which, under reasonable assumptions takes $\Omega(n^2/\delta^p)$ time. In the extreme case, when $p =\infty$, any approximate solver must transport all the mass in an approximately minimum-cost way which requires $\Omega(n^{2.5})$ time. In essence, the sensitivity to noise is precisely why additive approximations for the $p$-Wasserstein distance require $\Omega(n^2/\delta^p)$ time.
>
> In contrast, $p$-RPW is more robust, as a noise of $\delta$ impacts it by at most $\delta$. Our algorithm essentially transports all but $\delta$ mass in an approximate minimum-cost way and has an execution time of $O(n^2/\delta^3)$. Thus, it admits faster algorithms precisely because of its robustness to noise.
>
> ----
> ----
>
> >In line 207, lev(a,b) is never explicitly defined, I don't think. It is easy enough to infer, but please clarify.
>
> **Response:** The notation is defined in line 186.
>
>
> ----
> ----
>
> >I don't think I understood the second inequality in the derivation 181-189 (second column, for inequality 1). It was my impression that the telescoping sum would eliminate $H_{k^a_0}$ and $H_{k^b_0}$ if $h \geq 2$. I believe the result still holds, but would have expected different terms. Please clarify.
>
>
> **Response:** Thanks for raising this point. While the result of the equation in lines 181-189 is correct, the subscript of H in the first line of the equation has to be changed from $j-2$ to $j+2$. Note that in our notation, the root node (the largest cell) is at level 0. The slight typo resulted since the original paper presenting the HST construction [1] had the root at level $h$. We will change $j-2$ to $j+2$ in our next version.
>
> [1] J. Fakcharoenphol, S. Rao, and K. Talwar. "A tight bound on approximating arbitrary metrics by tree metrics." STOC, 2003.

---

### Official Review · Reviewer_NUZa · 2025-03-13

**Overall Recommendation:** 4

**Summary:**

This paper develops efficient algorithms for approximating the $p$-Wasserstein distance $W_p$ and a robust variant of $W_p$. In particular, when the input measures are uniform over discrete point sets of size $n$, they provide an approximation algorithm for $W_p$ with relative error $O(\log n)$ that runs in time $O(n^2 \log \Delta)$, where $\Delta$ is the ratio between largest and largest edge cost. Previous results of this form only worked with $p=1$.

For additive error, they provide a reduction suggesting that getting error $\delta$ requires time $O(n^2/\delta^p)$ unless one resorts to a class of currently impractical algorithms. Then, they show that a robust variant of the $W_p$ can be estimated in time $O(n^2/\delta^3)$ for all $p$.

## Update after Rebuttal
I maintain my positive evaluation.

**Claims And Evidence:**

Yes. This is primarily a theoretical paper and all theorems are accompanied by proofs which appear sound. They supported their guarantees with some basic experiments, with results that match or beat those predicted by the theory (which I think are sufficient given theoretical nature of the paper).

**Essential References Not Discussed:**

Not that I know of.

**Experimental Designs Or Analyses:**

The experiment setup seem fine as above. The results are in line (or even a bit better) than their theory predicts. By the way, can the authors elaborate on the $n^{3/2}$ scaling they observe - in particular, do they think there is room for a tighter computational complexity bound?

Can the authors please confirm that they will publish their code if the paper is accepted?

**Methods And Evaluation Criteria:**

Yes, I think the evaluation settings are fine. I assume they are from the setting where both sample sets are drawn from the same distribution (could be clarified). This makes sense because when the true distance is small the relative error is more tolerable.

**Other Comments Or Suggestions:**

Could you include the dependence of your complexities on p in the theorem statements? It seems to pretty well accounted for in the proofs, although I suppose there should be another multiplicative overhead of p when you perform arithmetic on entries of the exponentiated cost matrix (depending on the computational model).

Define LMR initialism

It seems that each cluster C is defined as a subset of points, so I am not sure what the notation X_C is needed for.

There seems to be a typo in the definition of $k_j^x$

I think the quantity $d(C)$ used in the statement of Lemma 2.3 is only defined in the proof.

**Other Strengths And Weaknesses:**

This paper is well-written, and I think that the first result and its proof ideas are quite nice - honestly enough to justify acceptance on their own. I could imagine that their tree metric might have other applications in computational geometry.

Since I am not as familiar with their robust variant, I am less sure of the significance of that result. E.g. it could be the case that it is easier to calculate for reasons that make it less useful in practice, though I am not claiming that to be the case.

**Questions For Authors:**

When p=1, how close is the robust W1 variant you consider to Dudley's bounded Lipschitz distance (IPM wrt Lipschitz and bounded functions)? That distance is also upper bounded by W1 and TV.

**Relation To Broader Scientific Literature:**

There is lots of work on statistics and computation of OT and its robust variants, which they discuss. I think they appropriately cite the relevant work on OT computation, though I work more on the statistical side so I cannot be certain. I am only loosely familiar with the data structure results that they use.

**Theoretical Claims:**

Yes, I read through all of the proofs for the relative approximation result and the reduction, and I read through the main parts of the additive approximation result (though I just skimmed over some auxiliary lemmas in the appendix). Everything looks good to me, assuming that they accurately describe the data structure results from previous work, which I am not very familiar with and did not verify.

There is a bit of ambiguity on the computational model. For example, reading the input points / computing the cost matrix takes $n^2 d$ time in $d$ dimensional Euclidean space. I think it's fine to primarily assume the distance matrix can be queried in constant time, but there should probably be a remark on this. Also, I am used to seeing additive approximation results which scale with the largest value of the cost matrix. I'm just confirming that there is not such a dependence here (assuming that $\Delta$ is small and that we can do arithmetic with entries of the distance matrix in constant time).

---

> ### Author Rebuttal · Authors · 2025-03-31
>
> We thank the reviewer for their thorough review and constructive feedback. We will update the paper to include your suggestions. We assume that distances can be queried in $O(1)$ time and will state it explicitly. We will also include additional dependencies of execution time on $p$ that arise due to taking the $p$th power of the costs.
>
> >Can the authors elaborate on the $n^{3/2}$ scaling they observe - in particular, do they think there is room for a tighter computational complexity bound?
>
> **Response:**
> Yes, it is possible to achieve a bound of $\tilde{O}(n^{3/2})$ for the special case of the minimum-cost bipartite matching problem provided the HSTs are precomputed. Sharathkumar and Agarwal [1] showed that Gabow and Tarjan's algorithm [2] for min-cost bipartite matching can be implemented using $\tilde{O}(n^{3/2})$ queries to the bichromatic closest pair data structure. Combining this with the structure of Section 2, we can bound the execution time by $\tilde{O}(n^{3/2})$ time algorithm. We will include this discussion in the next version of our paper.
>
> [1] R. Sharathkumar and P. K. Agarwal. "Algorithms for the transportation problem in geometric settings." SODA, 2012.
>
> [2] H. Gabow and R. E. Tarjan. "Faster scaling algorithms for network problems." SIAM Journal on Computing 1989.
>
> ----
> ----
>
> >Since I am not as familiar with their robust variant, I am less sure of the significance of that result. E.g. it could be the case that it is easier to calculate for reasons that make it less useful in practice, though I am not claiming that to be the case.
>
> **Response:** The $p$-Wasserstein distance's sensitivity to noise is a key factor limiting its practical utility and making the design of an additive approximation particularly challenging.
>
> A small noisy mass of $\delta^p$ can affect the $p$-Wasserstein distance by $\delta$. Consequently, any $\delta$-additive approximation algorithm must transport all but $\delta^p$ mass in an approximately minimum-cost way to ensure a bound of $\delta$ on the additive approximation, which, under reasonable assumptions takes $\Omega(n^2/\delta^p)$ time. In the extreme case, when $p =\infty$, any approximate solver must transport all the mass in an approximately minimum-cost way which requires $\Omega(n^{2.5})$ time. In essence, the sensitivity to noise is precisely why additive approximations for the $p$-Wasserstein distance require $\Omega(n^2/\delta^p)$ time.
>
> In contrast, $p$-RPW is more robust, as a noise of $\delta$ impacts it by at most $\delta$. Our algorithm essentially transports all but $\delta$ mass in an approximate minimum-cost way and has an execution time of $O(n^2/\delta^3)$. Therefore, the $p$-RPW distance is not only robust to noise but also allows for faster algorithms, a direct consequence of this robustness.
>
> ----
> ----
>
> >Can the authors please confirm that they will publish their code if the paper is accepted?
>
> **Response:** Yes, we will make our GitHub repository public.
>
> ----
> ----
>
>
> >When $p=1$, how close is the robust W1 variant you consider to Dudley's bounded Lipschitz distance (IPM w.r.t Lipschitz and bounded functions)?
>
> **Response:** For $p = 1$ and in a metric space with a unit diameter, the $1$-RPW can be written as $\sup_{Lip(f) \le 1-||f||_\infty} |\int f d\mu - \int f d\nu|$, which is upper-bounded by Dudley's bounded Lipschitz distance.

---

> > ### Comment · Reviewer_NUZa · 2025-04-01
> >
> > Thanks for your detailed response. I mostly buy the second response, although I imagine there are settings where the sensitivity is a feature. In any case, I do not object to RPW being useful and worth studying. The final connection to Dudley's metric is nice. I maintain my positive score.

---

### Official Review · Reviewer_wEzT · 2025-03-14

**Overall Recommendation:** 3

**Summary:**

This paper is concerned with developing a new algorithm for estimating p-Wasserstein distances. Notably, this algorithm enables approximating  the $p$-Wasserstein distance between distributions supported on $n$ atoms up to a multiplicative factor which scales as $O(log(n)$ (in expectation), i.e. the approximation satisfies $0\leq \mathbb E[A(\mu,\nu)]\leq C\log(n) W_p(\mu,\nu)$. The approximation can be computed in time scaling as $O(n^2\log n \log U\log \Delta)$, where $U,\Delta$ are problem dependent parameters. The approach is based on using a number of hierarchically well-separated trees which are constructed independently and computing a "tree-based distance" for each instance. This distance can be computed efficiently using a dynamic bichromatic closest pair (BCP) data structure which can be used to compute Wasserstein distances exactly in $O(n^2\Phi(n)\log(U))$ time, where $\Phi(n)$ is the query/update time for the underlying data structure (in this implementation, $O(\log(\Delta)\log(n))$).

It is then argued that a $\delta$-additive approximation of the $p$-Wasserstein distance with an execution time of $O(n^2/\delta^{p(1-\epsilon)})$ is unlikely to exist. On the other hand, by adapting a version of the LMR algorithm algorithm, it is shown that a $\delta$-additive approximation of the so-called RPW problem can be obtained in $O(n^2/\delta^3)$ time. The paper concludes with some experimental validations of these findings.

## update after rebuttal

The author's rebuttal addressed my primary question with this work, I have updated my score in consequence.

**Claims And Evidence:**

The paper claims to provide an improved algorithm for approximating the $p$-Wasserstein distance between distributions supported on $n$ atoms up to a multiplicative factor which scales as $O(log(n)$ (in expectation), i.e. the approximation satisfies $0\leq \mathbb E[A(\mu,\nu)]\leq C\log(n) W_p(\mu,\nu)$ and that this approximation can be computed in time scaling as $O(n^2\log n \log U\log \Delta)$. The article carefully explains the ideas underlying this approximation and the mathematical results are supported by proofs provided in the supplement. As such, I have no concerns regarding the accuracy of the claims made in the paper.

The claims regarding the RPW problem and $\delta$-additive approximations are also well supported and appear reasonable to me.

These claims are also empirically validated in numerical experiments which I find to be sufficient. Notably, the multiplicative scaling factor in the approximation is seen to scale sublinearly.

**Essential References Not Discussed:**

N/A

**Experimental Designs Or Analyses:**

I believe the experiments are sound.

**Methods And Evaluation Criteria:**

While the experiments are not extensive, they adequately demonstrate the claimed results.

**Other Comments Or Suggestions:**

Line 188 left column: distace -> distance

**Other Strengths And Weaknesses:**

I believe the paper is quite well-written and provides clear explanations. The algorithmic aspects of the work are particularly well explained. The algorithm for the RPW distance is also a nice contribution and its scaling appears favorable. My concern lies with the quality of the approximation provided for the Wasserstein distance. In effect, a  $O(\log n)$ multiplicative factor in the quality of the approximation appears very undesirable even if it is more efficient to compute. Although the experiments demonstrate that in two simple examples the dependence on $n$ is not bad, the approximation error is on the order of 1.5, which implies that this value is not a very good estimate for the actual Wasserstein distance.

**Questions For Authors:**

My only question pertains to the $O(\log n)$ multiplicative factor in the approximation of the Wasserstein distance. Perhaps I have misunderstood something, but this appears to be be a very serious limitation of the approach and implies that the quality of the approximation is effectively unknown.

**Relation To Broader Scientific Literature:**

The paper advances our understanding of algorithms for numerical resolutions of optimal transport and furnishes an efficient algorithm for estimating the RPW problem (based on a modification of a known algorithm).

**Theoretical Claims:**

I did not verify the proofs in the supplement.

---

> ### Author Rebuttal · Authors · 2025-03-31
>
> We thank the reviewer for their thorough review and constructive feedback. We answer the main concern raised by the reviewer below.
>
> >The importance of $O(\log n)$ approximation factor.
>
> **Response:** A tree-embedding-based $O(\log n)$-approximation for the $1$-Wasserstein distance originally developed by [1, 2] has been effectively utilized in various applications. These applications include the design of the FlowTree algorithm [3], $k$-NN and LSH data structures [4, 5], $1$-Wasserstein barycenter computation [6, 7], streaming algorithms for $1$-Wasserstein [8], design of $(1+\varepsilon)$-relative approximation algorithms [9, 10] for the $1$-Wasserstein distance, improving the accuracy of additive approximations such as Sinkhorn for the $1$-Wasserstein distance [11], and defining related metrics such as the tree-sliced Wasserstein distance as a proxy for the $1$-Wasserstein distance [12]. For many of these applications, a bound of $O(\log n)$ on the approximation factor is critical.
>
>
> Despite considerable efforts including some lower bounds [13], there are very few known approximation algorithms for the $p$-Wasserstein distance for $p > 1$ [14, 15]. In this paper, we introduce the **first $O(\log n)$-approximation algorithm** for computing the $p$-Wasserstein distance for any fixed $p \ge 1$. Extending our algorithm to applications such as $k$-NN under the $2$-Wasserstein distance or boosting the accuracy to $(1+\varepsilon)$-approximation remains an important future direction of research.
>
> ----
>
> [1] M. S. Charikar. "Similarity estimation techniques from rounding algorithms". STOC, 2002.
>
>
> [2] J. Kleinberg and E. Tardos. "Approximation algorithms for classification problems with pairwise relationships: Metric labeling and Markov random fields." JACM, 2002.
>
> [3] A. Backurs, Y. Dong, P. Indyk, I. Razenshteyn, and T. Wagner. "Scalable nearest neighbor search for optimal transport." ICML, 2020.
>
> [4] P. Indyk and N. Thaper. "Fast image retrieval via embeddings". International Workshop on Statistical and Computational Theories of Vision, 2003.
>
> [5] T. Liu, A. Moore, K. Yang, and A. Gray. "An investigation of practical approximate nearest neighbor algorithms." NeurIPS 2004.
>
> [6] T. Le, V. Huynh, N. Ho, D. Phung, and M. Yamada. "Tree-Wasserstein barycenter for large-scale multilevel clustering and scalable Bayes." ArXiv:1910.04483, 2019.
>
> [7] P. K. Agarwal, S. Raghvendra, P. Shirzadian, and K. Yao. "Efficient Approximation Algorithm for Computing Wasserstein Barycenter under Euclidean Metric." SODA, 2025.
>
> [8] X. Chen, R. Jayaram, A. Levi, and E. Waingarten. "New streaming algorithms for high dimensional EMD and MST." STOC, 2022.
>
> [9] J. Sherman. "Generalized preconditioning and undirected minimum-cost flow." SODA, 2017.
>
> [10] P. K. Agarwal, S. Raghvendra, P. Shirzadian, and K. Yao. "Fast and accurate approximations of the optimal transport in semi-discrete and discrete settings." SODA, 2024.
>
> [11] P. K. Agarwal, S. Raghvendra, P. Shirzadian, and R. Sowle. "A higher precision algorithm for computing the 1-Wasserstein distance." ICLR, 2023.
>
>
> [12] T. Le, M. Yamada, K. Fukumizu, and M. Cuturi. "Tree-sliced variants of Wasserstein distances." NeurIPS 2019.
>
> [13] A. Andoni, A. Naor, and O. Neiman. "Impossibility of sketching of the 3d transportation metric with quadratic cost." ICALP, 2016.
>
> [14] N. Lahn and S. Raghvendra. "An $O(n^{5/4})$ Time $\varepsilon$-Approximation Algorithm for RMS Matching in a Plane." SODA, 2021.
>
> [15] P. K. Agarwal and J. M. Phillips. "On bipartite matching under the RMS distance." CCCG 2006.

---

### Decision · Program_Chairs · 2025-05-01

**Decision:**

Accept (poster)

**Comment:**

The authors propose an $\mathcal{O}(\log{n})$-approximation algorithm for $p$-Wasserstein ($p \ge 2$) for discrete measures in time complexity $\mathcal{O}(n^2 \log{n})$, where $n$ is the number of supports of input measures. Intuitively, the proposed method is based on the construction of Hierarchically Well-Separated Trees (Fakcharoenphol et al., 2003) and the Hungarian algorithm with Dynamic Bichromatic Closest Pairs data structure. Additionally, the authors propose to approximate a robust variant of $p$-Wasserstein (i.e., $p$-PRW) within an additive error $\delta$ in time complexity $\mathcal{O}(n^2 / \delta^3)$, in contrast to $\Omega(n^2 / \delta^p)$ for existing combinatorial approaches in the literature. All Reviewers agree that it is a valuable contribution. The Reviewers raised concerns on empirical comparison to other popular approaches in the literature to illustrate the practical benefits of the proposed approach, i.e., whether one can realize an efficient implementation of the proposed approach to match with its theoretical complexity. We suggest the authors to incorporate the Reviewers' comments and discussions in the rebuttal into the updated version, which helps to strengthen the work.